# First Attentions Last: Better Exploiting First Attentions for Efficient Transformer Training

**Gyudong Kim**[1] **Hyukju Na**[1] **Jin Hyeon Kim**[1] **Hyunsung Jang**[2] **Jaemin Park**[2]
**Jaegi Hwang**[2] **Namkoo Ha**[2] **Seungryong Kim**[3†] **Young Geun Kim**[1†]
[1]Korea University [2]LIG Nex1 Co., Ltd. [3]KAIST AI

## Abstract

As training billion-scale transformers becomes increasingly common, employing multiple distributed GPUs along with parallel training methods has become a standard practice. However, existing transformer designs suffer from significant communication overhead, especially in Tensor Parallelism (TP), where each block's MHA–MLP connection requires an all-reduce communication. Through our investigation, we show that the MHA-MLP connections can be bypassed for efficiency, while the attention output of the first layer can serve as an alternative signal for the bypassed connection. Motivated by the observations, we propose **FAL** (*First Attentions Last*), an efficient transformer architecture that redirects the first MHA output to the MLP inputs of the following layers, eliminating the per-block MHA-MLP connections. This removes the all-reduce communication and enables parallel execution of MHA and MLP on a single GPU. We also introduce **FAL+**, which adds the normalized first attention output to the MHA outputs of the following layers to augment the MLP input for the model quality. Our evaluation shows that FAL reduces multi-GPU training time by up to 44%, improves single-GPU throughput by up to 1.18×, and achieves better perplexity compared to the baseline GPT. FAL+ achieves even lower perplexity without increasing the training time than the baseline. **Codes are available at: https://casl-ku.github.io/FAL/**

## 1 Introduction

As transformers continue to grow following scaling law trends [1], Large Language Models (LLMs) such as GPT [2] and LLaMA [3] demonstrate superior performance across a wide range of NLP tasks. Given that the large transformers usually have billions of parameters which is far exceeding a single GPU's memory and computation limit, distributed training over multiple GPUs is necessary.

Among distributed training methods, Tensor Parallelism (TP) [4] has been considered as a standard practice for multi-GPU training in each single-node, given its decent computation throughput and memory efficiency [5, 6, 7]. However, the overall efficiency of TP can still be limited by the communication overhead across the GPUs. For example, if the number of GPUs increases (and/or the interconnect speed slows down), the efficiency of TP can be significantly degraded due to the increased communication overhead. Hence, to further scale up transformers with more GPUs and diverse interconnect configurations, it is essential to mitigate the communication overhead in TP.

One of the major sources of communication overhead in TP is the per-block communication (i.e., all-reduce) required across GPUs to transfer data between two main modules in each transformer block: Multi Head Attention (MHA) and Multi Layer Perceptron (MLP). In TP, each GPU computes a part of the MHA output, but the results must be aggregated via all-reduce to form a complete activation before being passed to the MLP.

---

† Corresponding authors (seungryong.kim@kaist.ac.kr, younggeun_kim@korea.ac.kr)

39th Conference on Neural Information Processing Systems (NeurIPS 2025).

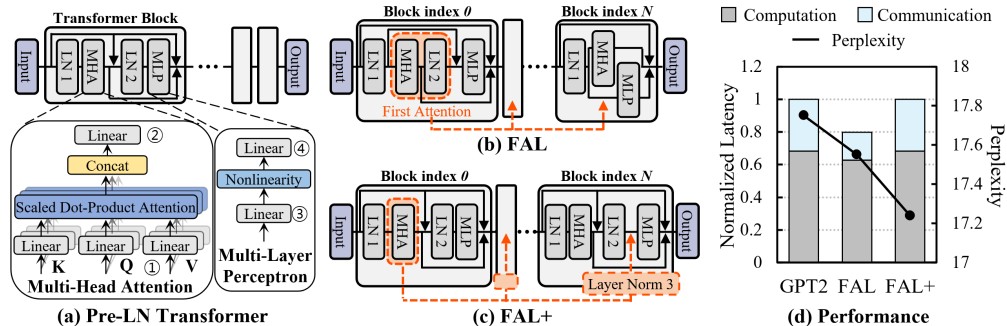

Figure 1: **Transformer block designs highlighting our proposed modifications.** (a) Pre-LN architecture with Layer Normalization (LN), Multi Head Attention (MHA) and Multi Layer Perceptron (MLP). (b) FAL blocks with reconfigured connections. (c) FAL+ blocks with augmented connections. (d) End-to-end training time and perplexity comparison of Pre-LN architecture, FAL, and FAL+.

To reduce communication overhead in TP, we question whether the direct connections between two main modules in each transformer block (i.e., MHA-MLP connections) are strictly necessary. To answer this question, we conduct a series of analyses (Sec. 3), revealing two key observations:

1. **MHA-MLP Connections Can Be Reconfigured**: We find that the direct MHA-MLP connection is not always essential, as the residual path already accumulates MHA outputs from prior blocks. However, naively removing the connection causes a large drop in model quality, indicating additional mechanisms are required to mitigate information loss.
2. **First Attention is Key**: Our analysis shows that the first MHA output has a disproportionately large impact on final predictions. Thus, leveraging the first attention more effectively can compensate for the information loss caused by the removed connections.

Motivated by our findings, we propose **FAL** (*First Attentions Last*), an efficient parallel transformer architecture that redirects the high-impact first MHA output to the MLP inputs of following blocks, rather than relying on the attention within each block (Fig. 1 (b)). This modification eliminates expensive all-reduce communication within each block and also enables parallel execution of MHA and MLP on a single GPU. As a result, FAL improves training throughput by 1.07-1.52× in multi-GPU settings, and by 1.03-1.18× on a single GPU, compared to the standard transformer. Moreover, carefully leveraging the high-impact first attention not only preserves model quality but improves it, reducing validation perplexity and increasing the average SuperGLUE [8] score. To further improve the model quality leveraging the high-impact first attention, we additionally propose **FAL+**, a variant of FAL, which augments MHA–MLP connections rather than removing them (Fig. 1 (c)). FAL+ achieves even lower perplexity than FAL, further confirming the importance of the first attention.

## 2 Background

**Transformer Architecture.** A transformer model consists of a stack of transformer blocks. Fig. 1 (a) depicts a diagram of Pre-LayerNorm (Pre-LN) architecture, which is widely adopted for deep transformers due to its superior training stability and signal propagation [9, 10, 11]. Each transformer block includes Multi Head Attention (MHA) for dependency modeling and a Multi Layer Perceptron (MLP) for position-wise transformation using the dependencies computed by MHA.

The MHA first projects the input into multiple heads via query (Q), key (K), and value (V) layers (①). Each head independently learns dependencies through scaled dot-product attention. The outputs from all heads are concatenated and merged by a linear layer (②). The MLP then processes the MHA output using position-wise transformations. It first projects the MHA output to a higher dimensional space (③), applies a GeLU (or ReLU) non-linearity, and then projects it back to its original dimension (④). As a result, given the block input $X_i$, the output of the $i$-th Transformer block can be formulated as:

$$X_i + \mathrm{MHA}_i(\mathrm{LN}(X_i)) + \mathrm{MLP}_i(\mathrm{LN}(X_i + \mathrm{MHA}_i(\mathrm{LN}(X_i)))) \qquad (1)$$

Here, $X_i$ is progressively refined by the MHA and MLP, supported by residual connections [12] and layer normalization (LN) [13]. These residual connections and layer normalizations are essential

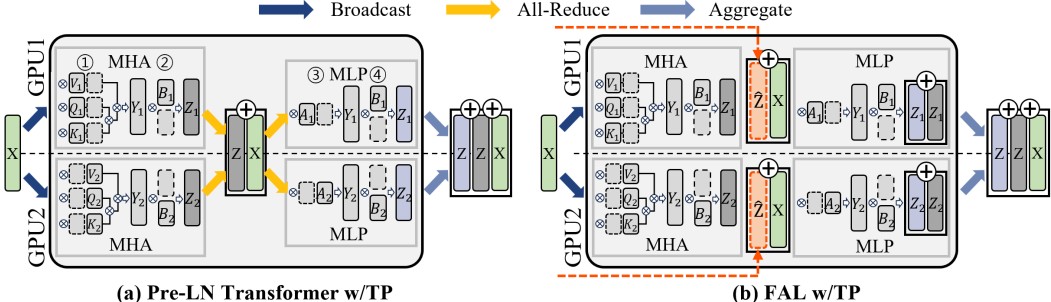

Figure 2: **Forward pass during Transformer training with tensor parallelism across 2 GPUs.** (a) Standard training involves Broadcast, All-Reduce, and Aggregate communication. (b) FAL training requires only Broadcast and Aggregate, reducing communication overhead.

for stabilizing gradients and preventing rank collapse [14]. Pre-LN design forms the basis for many large-scale models such as GPT [2] and LLaMA [3]. In Sec. 4, we build upon this baseline to propose structural modifications for efficiency and model quality.

**Tensor Parallelism for Transformer.** Fig. 2 (a) illustrates tensor parallelism (TP) [4], which is one of the widely used parallel training methods. In TP, the parameters of the transformer are sliced either along the row (input) or column (output) dimensions and distributed across GPUs.

The input is first broadcasted across GPUs and then fed into the sliced MHA. The outputs of the sliced MHA are then summed up together to be passed to the subsequent MLP. This results in all-reduce communication between GPUs to fully assemble the MHA output before passing it to the MLP. After that, the assembled MHA output is fed into the sliced MLP. The outputs of the sliced MLP also need to be summed up together to be passed to the next transformer block — this requires an aggregate-broadcast communication.

Unlike data parallelism [15] and pipeline parallelism [16], which require moving model states or introducing bubble periods, TP can achieve high computational occupancy when the model size is much larger than the activation footprint [7].[1] However, it still suffers from substantial communication overhead as the number of GPUs increases or the interconnect speed slows down [5, 17]. Since TP partitions MHA and MLP parameters across GPUs, each transformer block requires two all-reduces in both forward and backward passes, creating significant overhead. To mitigate this, we focus on reducing the necessity of all-reduce within each transformer block.

## 3   Motivation

A crucial principle in typical transformer architectures is that the MLP must always receive the most recent MHA output within each block. This incurs all-reduce communication in TP affecting the overall training time. Here, given that the residual path in a specific block already accumulates the attention outputs of all preceding blocks, we question whether the MLP truly needs the most recent MHA output. To answer the question, we conduct several analyses on GPT-2, using four linguistic datasets — WikiText-2 [18], PTB [19], BookCorpus [20], and CC-News [21].

### 3.1   MHA-MLP Connections Can Be Reconfigured

To examine whether the direct MHA-MLP connection is strictly necessary, we first measure how much a single MHA output has an impact on the MLP input within a block. Specifically, we quantify the feature representation changes between the MHA output and MLP input by comparing their CKA similarity [22] — similar practice is used in many prior works to quantify the feature representation changes across activations in the model [23, 24].

Fig. 3 (a) shows the CKA similarity scores for MHA outputs, MLP inputs (*Residual + MHA*), and MLP outputs across adjacent blocks. Although the MHA output, which largely varies across the

---

[1]A comparison of training times among data, pipeline, and tensor parallelism is provided in Apdx. B.

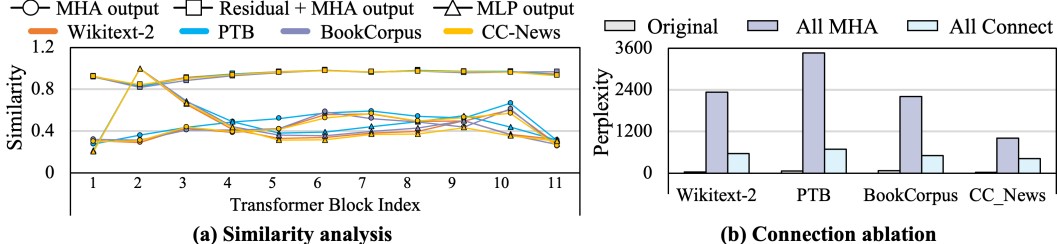

**(a) Similarity analysis**

**(b) Connection ablation**

Figure 3: **Motivation for reconfiguring MHA-MLP connections.** (a) CKA similarity analysis across successive GPT-2 blocks. The x-axis indicates the block index, and the y-axis shows the similarity between consecutive MHA outputs, Residual+MHA outputs (i.e., MLP inputs), and MLP outputs. (b) Connection ablation results measured by perplexity. *Original* denotes the unaltered model, *All MHA* removes all MHA layer, and *All Connect* removes all direct MHA-MLP connections.

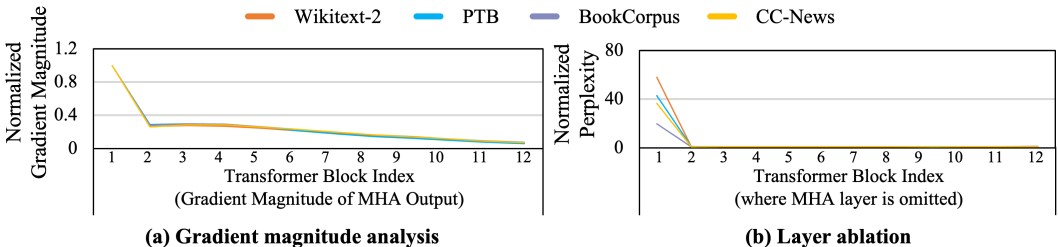

**(a) Gradient magnitude analysis**

**(b) Layer ablation**

Figure 4: **The crucial role of the first attentions.** (a) Normalized gradient magnitude of the MHA outputs across Transformer blocks in GPT-2 for different datasets. The x-axis represents the block index. (b) Layer ablation results measured by perplexity. The x-axis indicates the index of the transformer block from which the MHA is omitted.

blocks, is added into the MLP input, the MLP input does not vary much across the blocks. This suggests that a single MHA output change has a limited impact on the MLP inputs since the residual connection already accumulates attention signals from the previous blocks. Note that the MLP still produces different outputs, meaning that new information can be generated even if the inputs are similar. This implies that the MLP may not always require the most recent MHA output (i.e., the output of the MHA within the same block).

Given the observation that MHA output and MLP input within the same block consistently yields similar representations, we hypothesize that the MHA-MLP connections can be selectively skipped or reconfigured for efficiency gain — high representational similarity of intermediate features often leaves potential for lightweight modifications or pruning [23]. To validate this hypothesis, we remove either entire MHAs or only their connections to the MLP and quantify the impact of the removals.

Fig. 3 (b) illustrates two scenarios: removing all MHAs (*All MHA*) versus removing all MHA-MLP connections (*All Connect*). As expected, removing *All MHA* severely degrades model quality, since no attention can be utilized at all. In contrast, removing *All Connect* recovers much of the lost performance compared to removing the entire MHAs. Although skipping connections is less harmful than removing entire layers, the performance loss remains significant, indicating the need for alternative signals for MLP.

## 3.2 First Attention is Key

To investigate alternative attention signals for MLP, we measure the gradient magnitude of the MHA outputs across all blocks to find a crucial attention output with a high impact on final predictions. Note gradient magnitude is also one of the widely used method to measure which input features the model is focusing on as an importance score [25].

Fig. 4 (a) shows that the first MHA output consistently exhibits the highest gradient magnitude, indicating that perturbations in the earliest attention result have a disproportionately large impact on the final prediction. We further confirm this by measuring the perplexity after omitting the

MHA from individual transformer blocks. As shown in Fig. 4 (b), removing the first attention causes a far larger perplexity increase than removing later layers, verifying the crucial role of the first attention in language modeling. These findings align with the well-known psychological phenomenon of the primacy effect [26], commonly summarized as "first impressions matter." The primacy effect of the first attention is not limited to a specific model architecture — previous works also identified the prominent impact of the first attention layer across various attention mechanisms and tasks [27, 28, 29].

Our findings suggest that leveraging the first attention more effectively can compensate for the information loss caused by skipping direct MHA-MLP connections.[2]

## 4 FAL: Harnessing First Attention for Enhanced Efficiency

Our analyses in Sec. 3 indicate 1) although MHA outputs can be bypassed, alternative signals for MLP are needed (Sec. 3.1), and 2) the first MHA output could be the key to bridging the performance gap (Sec. 3.2). Building on these findings, we propose **FAL** (First Attentions Last), a novel transformer architecture designed to streamline MHA-MLP connections using the first MHA. We emphasize three key considerations to balance efficiency and model quality:

- **Retaining Crucial First-Attention Signal:** Motivated by dual process theory [30] which suggests that revisiting the initial judgement can improve accuracy, FAL carefully retains (and dwells on) the crucial first attention signal in the subsequent blocks, by distinctly treating the first block as a specialized "preparation" stage.
- **Maintaining the Transformer Structure:** Apart from rerouting the MLP input, FAL retains the conventional ordering of sub-layers, residual paths, and multi head attention components. Our design thus preserves the proven benefits of standard residual connections and MHA while enabling more careful use of the first attention output.
- **Minimizing Overhead:** The reconfiguration must reduce data communication in multi-GPU training while avoiding excessive memory usage or computations.

### 4.1 FAL Architecture

Fig. 1 (b) illustrates the core design of FAL. FAL revisits the first attention output in all subsequent blocks, reducing the need to rely on the most recent attention output at each block. For reuse, we apply layer normalization (LN) to the first attention output only once in the first block, thereby maintaining a stable scale without incurring repeated LN overhead. Because the first attention output is passed through every block, FAL acts akin to a dense skip connection [31] for this highly influential first-stage signal.

The MLP now receives $\text{LN}(X_i) + \text{LN}(\text{MHA}_1(\text{LN}(X_1)))$, instead of $\text{LN}(X_i + \text{MHA}_i(\text{LN}(X_i)))$. By adding these two LN outputs together, **LN**$(\text{MHA}_1(\text{LN}(X_1)))$ and **LN**$(X_i)$, our design makes the *LN affine parameters* learn the relative weight of each component. [3] Meanwhile, each block still computes its own MHA (i.e., $\text{MHA}_i$), which remains in the residual connection at the output of the block to incorporate newly captured information. As a result, the $i$th FAL block output is formulated as:

$$X_i + \text{MHA}_i(\text{LN}(X_i)) + \text{MLP}_i(\text{LN}(X_i) + \text{LN}(\text{MHA}_1(\text{LN}(X_1)))) \tag{2}$$

### 4.2 Effectiveness of FAL

**Reducing Communication in Multi-GPU Parallelism.** With reconfigured connections, FAL lowers communication overhead during multi-GPU training. Fig. 2 (b) illustrates FAL training using TP, which fuses the all-reduce operation for MHA into that for MLP. By connecting the first MHA, instead of the latest one, to the MLP, the MLP no longer requires the all-reduce output. As a result, the outputs of the latest MHA and the MLP can be added locally on each GPU. This halves the communication overhead, significantly improving the training time performance.

---

[2]Additional analyses (Apdx. C) consistently support (1) the feasibility of bypassing per-block MHA outputs, and (2) the pivotal role of the first MHA output across transformer variants and domains.

[3]To avoid storing extra activations, we reposition the first layer's LN from the MLP input to the MHA output. This allows later blocks to reuse the normalized first-attention output without recomputation or extra memory, as LN result is already cached during the forward pass for backpropagation.

**Single-GPU Acceleration.** FAL also speeds up single-GPU training. When executing a standard transformer block on a single GPU, the MLP must wait until the MHA finishes its operations. This sequential process can lead to sub-optimal utilization of computational resources during training [32].

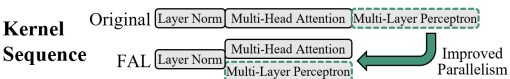

Figure 5: **Parallel kernel launch of FAL.**

As depicted in Figure 5, because FAL modules (i.e., MHA and MLP) have no dependencies, they can be executed in parallel accelerating single-GPU training.[4] For example, while memory-intensive operations (e.g., element-wise operations) of a module are executed, compute-intensive operations (e.g., matrix multiplications) of the other module can be simultaneously executed. This concurrency increases both computational and memory throughput, reducing overall training time.

Note, although we focus on the training efficiency in this paper, the benefits of FAL can also be applied to multi-GPU inference scenarios — TP is widely used even for the multi-GPU execution of inference due to its decent computation throughput and memory efficiency [7]. The detailed results of FAL's inference acceleration are presented in Apdx. D.3.

# 5 FAL+: Harnessing First Attention for Enhanced Quality

Our observations in Sec. 3.2 highlight the crucial role of the first MHA in language modeling. In FAL, we exploit the first MHA output to reduce overhead by replacing subsequent direct MHA–MLP connections. In this section, we introduce **FAL+**, which instead augments the original MHA-MLP connections with the first MHA output to further enhance model quality.

Fig. 1 (c) illustrates the core design of FAL+. Rather than removing any MHA-MLP connections, each transformer block integrates the first attention signal $MHA_1(LN(X_1))$ alongside its original connection. FAL+ appends an additional LN on each block for the first attention signal, allowing the LN affine parameters to control how much of the first attention is utilized.

As we show in Sec. 6.4, FAL+ consistently achieves lower perplexity than the baseline, demonstrating that the first MHA output can be leveraged effectively even when the primary goal is to improve model quality rather than training speed.

# 6 Evaluation Results and Analysis

## 6.1 Experimental Setup

We conduct our experiments on a variety of hardware, datasets, and models with various scales, as briefly summarized below (further details in Apdx. A).

- **Hardware:** In order to comprehensively evaluate our approach across diverse GPU architectures and scales, we conduct experiments on multi-GPU configurations (2–8 GPUs) with RTX 3090 and H200 devices connected via PCIe or NVLink, and on single-GPU setups with RTX 3090, RTX 4090, and RTX A6000.
- **Baselines:** We compare FAL and FAL+ with a standard transformer-based language model GPT-2 and larger GPT variants — FAL and FAL+ are implemented atop the baselines. We also compare FAL and FAL+ with a parallel configuration considered in [33, 34, 35] where MHA and MLP modules are executed simultaneously using the same input, in order to validate that reusing the first MHA output shows similar parallelism improvements with the parallel configuration while even enhancing the model quality.
  In multi-GPU scenarios, we further compare our proposed architectures with lossy communication time reduction methods, such as gradient quantization [36] and low rank approximation [37].
- **Datasets:** We pre-train the models on OpenWebText corpus [38], a publicly available counterpart to GPT-2's WebText. For scalability analysis, we use the Pile dataset [39]. Zero-shot performance is evaluated on language understanding tasks using the SuperGLUE benchmark suite [8].

---

[4]In practice, FAL enables overlapped execution of MHA and MLP on separate CUDA streams. Such overlap allows the warp scheduler to better hide latency — when one warp stalls on a memory operation, another ready warp in a different stream can be scheduled without delay.

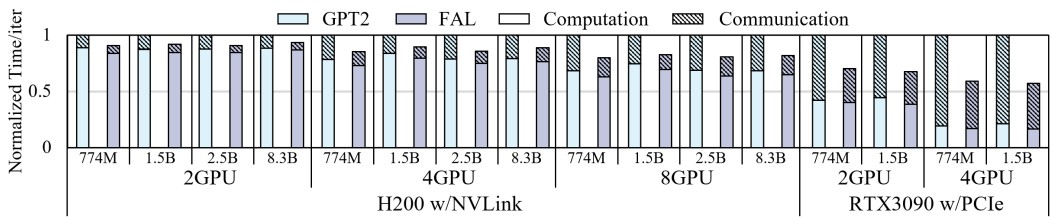

Figure 6: **Normalized Multi-GPU Training Time of GPT2 and FAL.**

## 6.2 Multi-GPU Performance

Fig. 6 shows the normalized training time of GPT-2 and FAL on multi-GPUs (i.e., H200 with NVLink and RTX 3090 with PCIe) with model sizes from 774M to 8.3B. For high-end GPUs with NVLink, FAL improves the training time performance by 13.2% on average (up to 20.1%) compared to GPT-2. In the case of 2 GPUs, communication overhead is relatively low due to the high bandwidth of NVLink. In this case, FAL's performance improvement mainly comes from the single-GPU acceleration. As the number of GPUs increases — for training larger models — the communication overhead substantially increases. In such cases, FAL further improves performance (by 18.7% on average) by reducing all-reduce communication within each transformer block.

In typical multi-GPU servers with PCIe (instead of NVLink) [40], the communication overhead becomes more pronounced accounting for up to 80.6% of training time on 4 GPUs. In such cases, FAL provides even greater benefit — it improves training time performance by 36.6% on average (up to 43.1%) compared to GPT-2. These results demonstrate that FAL is not only effective in high-performance setting, but also highly beneficial in typical PCIe-based settings, reinforcing its scalability and practicality across a variety of deployment settings.

**Comparison with Lossy Communication Reduction Methods.** Fig. 7 shows the training time breakdown and perplexity of GPT-2, FAL, and two gradient compression methods (quantization [36] and low-rank approximation [37] denoted as Grad-Q and Grad-LR, respectively) on a 2-GPU PCIe setup with OpenWebText. While the compression techniques substantially reduce communication time (by 37.8% on average), they significantly degrade model quality. On the other hand, FAL reduces much more communication time overhead (by 49.4%) compared to the compression techniques without compromising the model quality — it even reduces the perplexity compared to GPT-2. This result demonstrates that FAL can strike much better performance-accuracy trade-off point compared to the prior communication reduction techniques via the connection reconfiguration.

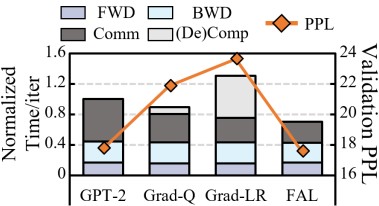

Figure 7: **Validation perplexity and training time breakdown of GPT-2, gradient compression methods, and FAL.** (FWD: forward, BWD: backward, Comm: communication, (De)Comp: compression/decompression time)

## 6.3 Single-GPU Performance

Fig. 8 (a) shows the normalized throughput (tokens per second) of GPT-2 and FAL on single GPUs (RTX3090, RTX4090, and RTX A6000); the throughput is normalized to that of GPT-2. As shown in Fig. 8 (a), FAL improves single-GPU throughput by 1.08× on average (up to 1.18×) compared to GPT-2. This is because FAL enables overlapped execution of MHA and MLP in each block better utilizing resources — Fig. 8 (b) shows that FAL improves SM utilization, warp occupancy, tensor core usage, and memory bandwidth by up to 8.2%, 45.9%, 13.9%, and 18.4%, respectively, on RTX-3090.

Note FAL typically shows better single GPU throughput when FlashAttention [41] is adopted. This is because FlashAttention increases the computational intensity of attentions with kernel fusion, giving more opportunity for FAL to overlap the computation-intensive operations and memory-intensive operations in MHA and MLP leading to better resource utilization. Although MLP's large matrix multiplies are compute-heavy, each GEMM begins and ends with global-memory loads and stores. These boundary memory transactions introduce unavoidable stalls, even if the core GEMM

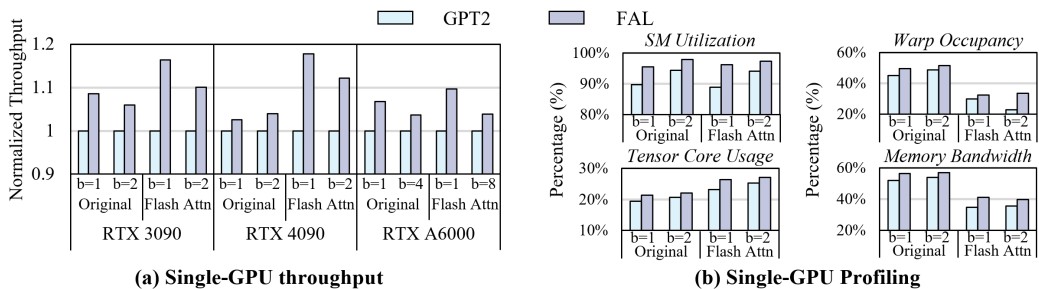

Figure 8: **Single-GPU throughput comparison between GPT2 and FAL.** (a) Normalized throughput (tokens/sec). (b) Analysis of throughput gains on RTX 3090.

Table 1: **Openwebtext validation perplexity, training time, and SuperGLUE zero-shot results.**

| | Openwebtext (↓) | SuperGLUE (↑) (CB, Record: F1 score, Others: Accuracy) | | | | | | | | |
|---|---|---|---|---|---|---|---|---|---|---|
| Model | PPL / Time | BoolQ [42] | CB [43] | COPA [44] | MultiRC [45] | ReCoRD [46] | RTE [47] | WIC [48] | WSC [49] | Avg. |
| GPT-2 774M | 17.75 / 13.2d | **55.7** | 19.4 | 54.0 | 52.3 | **57.4** | **54.2** | 49.8 | 45.2 | 48.5 |
| Parallel | 17.80 / **8.6d** | 50.0 | 19.4 | 58.0 | 53.8 | 48.6 | 51.6 | 49.1 | 36.5 | 45.9 |
| **FAL** | 17.55 / **8.6d** | 50.2 | **21.4** | **62.0** | 54.5 | 52.6 | 51.6 | 46.6 | **49.0** | 48.5 |
| **FAL+** | **17.24** / 13.2d | 51.8 | 21.1 | 58.0 | **55.7** | 56.2 | 51.3 | **51.3** | 48.1 | **49.2** |
| GPT-2 1.5B | 14.72 / 24.1d | 58.0 | 24.1 | 65.0 | **57.2** | 78.4 | 53.1 | **50.0** | 40.4 | 53.3 |
| **FAL** | 14.23 / **16.1d** | 58.1 | 21.6 | **72.0** | **57.2** | 78.7 | 54.2 | 49.2 | **64.4** | **56.9** |
| **FAL+** | **14.12** / 24.2d | **58.8** | 26.2 | 65.0 | **57.2** | **79.0** | **56.0** | 49.8 | 51.0 | 55.4 |

is compute-bound. FlashAttention's higher arithmetic intensity in the attention block lengthens its compute phase relative to its memory phase, creating more opportunities to hide MLP's boundary stalls behind attention computation.

## 6.4 Model Quality and Training Efficiency

**Training from Scratch.** To validate the model quality of FAL and FAL+, we measure the perplexity and end-to-end training time, respectively, while training GPT-2, parallel configuration (denoted as Parallel), FAL, and FAL+ with OpenWebText dataset on 4-GPU PCIe setup.

Table 1 (left) reports the perplexity and total training time for the 774M (36 layer) and 1.5B (48 layer) scales. Both FAL and Parallel improve the training time by 34%, on average, compared to GPT-2. However, Parallel degrades model quality compared to GPT-2, since it discards MHA-MLP connections without providing any alternative features. On the other hand, FAL even improves model quality — it lowers perplexity by 0.2 and 0.49, compared to GPT-2 774M and 1.5B, respectively. This result demonstrates that reconnecting the first MHA output with LN is not just an alternative signal of the removed connections, but a deliberate reuse of the crucial early representation which leads to better understanding of the input.

FAL+ achieves even lower perplexity compared to GPT-2, by augmenting the MHA-MLP connections with the first MHA output — its training time is thus almost same with the baseline. Here, the perplexity improvements of FAL and FAL+ get larger as model scale increases — this is mainly because the advantage of revisiting the first attention becomes increasingly effective in deeper models (see Scalability Analysis for more details).

**Scalability.** To validate the scalability of FAL and FAL+, we compare token ingestion efficiency of the Pre-LN Transformer, FAL, and FAL+ as their depth increases from 36 to 60 layers. Fig. 9 shows the loss curves under fast training conditions inspired by Cramming [50]. In the early stages of training, both FAL and FAL+ reduce MLM loss more rapidly than the Pre-LN baseline. In case of 36-layer scale, all models end up show similar loss values.

As the model depth increases, both FAL and FAL+ converge to lower MLM loss values than the Pre-LN baseline — FAL+ usually exhibits lower MLM loss values compared to FAL. In case of standard Pre-LN transformers, residual connections cannot fully preserve early-layer information, which is gradually diluted as activations accumulate through depth [9]. This limits deeper architectures from revisiting the high-impact first attention signal that strongly influences the model's final predictions. In contrast, FAL and FAL+ repeatedly reintroduce the high-impact first attention output across depth,

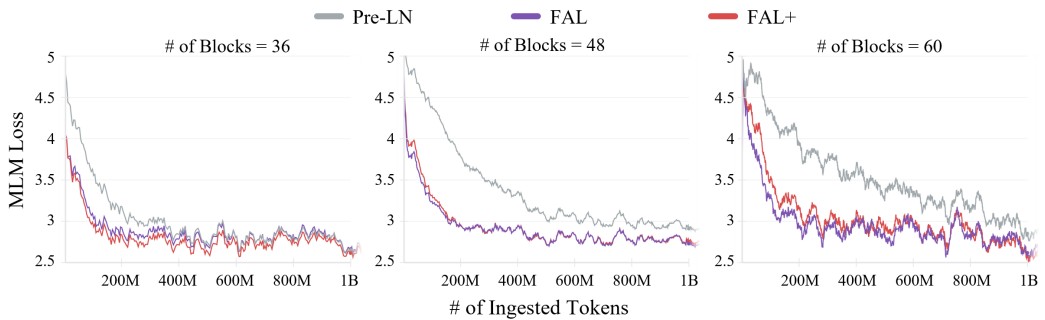

Figure 9: **Loss comparison with increasing number of blocks across Pre-LN architecture, FAL, and FAL+.**

allowing later blocks to adaptively re-weight this signal. Moreover, the degradation from removing direct MHA–MLP connections diminishes in deeper models, while the primacy of the first attention persists (as we demonstrate in Apdx. C). This validates the benefit of the replacement strategy of FAL in deeper architectures.

**Generalizability.**   To further validate the generalizability of FAL and FAL+, we evaluate their zero-shot performance on language understanding tasks using SuperGLUE benchmark. Table 1 (right) shows the zero-shot results, where CB and ReCoRD are evaluated using F1 score, while the remaining tasks use accuracy. FAL largely preserves the language modeling capabilities of the standard Pre-LN architecture. In the case of 774M scale, FAL achieves the same average score as GPT-2 across SuperGLUE tasks. On the other hand, in the case of 1.5B scale, FAL surpasses GPT-2 due to the increased depth which gives more opportunity to revisit the first attention signal. FAL+ achieves higher average scores than GPT-2 in both scales, by augmenting the MHA-MLP connections with the first MHA output. [5] Our motivational analyses on vision and multimodal tasks in Apdx. C also demonstrate the feasibility of bypassing per-block MHA outputs using the pivotal first MHA output. Hence, we believe FAL and FAL+ can further be generalized to other tasks.

We also validate FAL and FAL+ on various transformer variants, such as LLaMA (with Grouped Query Attention, GQA) and Switch Transformer (with Mixture of Experts, MoE). Both FAL and FAL+ consistently yield lower loss curves than the baseline GQA and MoE architectures, confirming that our connection-level reconfiguration is broadly applicable — the detailed results and training configurations are provided in Apdx. E.

**Downstream Robustness.**   We empirically examine how injecting the first attention signal during instruction tuning affects the trade-off between stability and adaptation. We fine-tune both GPT-2 1.5B and FAL+ 1.5B on the Alpaca instruction tuning dataset [53] using four learning rates (1e-5 to 1e-2).

Table 2 reports trained perplexity (Trained PPL) on Alpaca, reflecting adaptation, and validation perplexity degradation ($\Delta$ Val PPL) on Open-WebText, reflecting forgetting. FAL+ consistently preserves pretraining knowledge better than GPT-2. Across all learning rates (LR), FAL+ shows lower or equal $\Delta$ Val PPL on Open-

Table 2: **Instruction tuning robustness on Alpaca.** (stability vs. adaptation)

| Model | LR | $\Delta$ Val PPL | Trained PPL |
|---|---|---|---|
| GPT2 1.5B | 1e-5 | 0.01 | 30.93 |
| | 1e-4 | 0.00 | 30.92 |
| | 1e-3 | 0.01 | 28.10 |
| | 1e-2 | 1.39 | **4.83** |
| **FAL+** 1.5B | 1e-5 | **0.00** | 27.97 |
| | 1e-4 | **0.00** | 27.98 |
| | 1e-3 | **-0.01** | 27.22 |
| | 1e-2 | **0.61** | 5.76 |

---

[5]Although FAL+ exhibits lower perplexity compared to FAL during pretraining, its SuperGLUE score is lower than that of FAL. This discrepancy suggests that the domain mismatch between the OpenWebText corpus and SuperGLUE tasks may limit the direct transferability of perplexity improvements on the pretraining corpus to downstream performance [51, 52]. However, we expect this gap can be mitigated through continual pretraining on more diverse domain sources [51].

WebText compared to GPT-2, indicating stronger stability. At LR=1e-3, FAL+ even shows a slight improvement in validation PPL (-0.01), suggesting positive regularization rather than forgetting.

FAL+ also achieves better adaptation when forgetting is minimal. In Table 2, while maintaining zero or negative $\Delta$PPL, FAL+ attains the lowest trained PPL (27.22) at LR=1e-3, demonstrating high adaptability with minimal forgetting. In contrast, GPT-2 reaches a lower trained PPL (4.83) at LR=1e-2 only by severely compromising original domain knowledge ($\Delta$ Val PPL = 1.39). These results show that reusing the impactful first attention in FAL+ enables robust adaptation while mitigating catastrophic forgetting [54], ensuring that the improved model quality (Table 1) is not achieved at the cost of stability or downstream adaptability.

## 7  Related Work

**Parallel Training Methods:** Training large-scale neural networks requires parallel methods to handle high computation and memory demands. Data Parallelism (DP), Pipeline Parallelism (PP), and Tensor Parallelism (TP) are the primary approaches. In DP, each GPU processes different data batches with identical model replicas and synchronizes gradients, but scaling to large models incurs heavy memory and communication costs. PP [16] partitions (sub)layers into sequential stages across GPUs, introducing pipeline bubbles that slow training. TP [4] distributes model (sub)layers across GPUs to avoid bubbles, but communication still remains a bottleneck. FAL tackles this by exploiting the first attention to reduce communication while preserving information. Some works [5, 55] combine TP and PP with DP for efficient GPU training. FAL can further boost these combinations by lowering the communication cost introduced by TP.

**Communication Reduction in Parallel Training:** Overlapping communication with computation can hide communication delays [6], but its effectiveness depends on how much communication can be overlapped with the computation [56]. FAL offers more opportunity to overlap communication with computation by reducing the communication frequency. Gradient compression using quantization [36] or low-rank approximation [37] reduces data exchange, but the (de)compression overhead can negate time savings given TP's short communication intervals.

Parallel configurations within transformer blocks [33, 34, 35] lower TP communication [7] by using the same input to the MHA and MLP, yet degrade quality on linguistic tasks. In contrast, FAL reconfigures sequential connections using the first MHA output, thereby cutting communication overhead while improving model quality.

## 8  Conclusion

Training large transformer models over multiple GPUs often incurs substantial communication overhead. To alleviate the communication overhead, we propose **FAL** (*First Attentions Last*), a novel architecture that leverages the output of the high-impact first MHA to streamline MHA–MLP connections. Leveraging the first attention more effectively, FAL removes expensive all-reduce communication within each block and enables parallel execution of MHA and MLP on a single GPU without compromising the model quality. We also introduce **FAL+**, a variant that leverages the first attention output to pursue additional quality gains, underscoring the flexibility of our connection-level approach. In our evaluation across various linguistic tasks and hardware configurations, FAL shows up to 44% of training time reduction, compared to the baselines of GPT architecture, while improving model quality as the depth increases. Furthermore, **FAL+** achieves even better model quality with connection augmentation. We believe this connection-level perspective opens new avenues for refining transformer architectures, both in terms of communication efficiency and model quality.

**Limitation and Future Work.**   FAL slightly degrades accuracy for small models (0.3%, Apdx. E.2) because shallow networks provide fewer layers to accumulate attention signals, making the replacement of block-level connections less stable. FAL+, however, augments rather than replaces these connections, allowing later blocks to leverage both the current attention and the first attention, which avoids information loss and even yields a slight accuracy gain (0.14%). Building upon this, incorporating a connection-reconfiguration strategy centered on the first attention output into Neural Architecture Search [57] or dynamically injecting the first attention through gating [58] presents an interesting direction for future research.

## Acknowledgements

This work was supported in part by National Research Foundation of Korea (NRF) grant funded by the Korea government (MSIT) (RS-2023-00212711, RS-2025-24534857, and RS-2025-25434746), Institute of Information & Communications Technology Planning & Evaluation (IITP) - ITRC (Information Technology Research Center) grant funded by the MSIT (IITP-2025-RS-2023-00260091), and ICT Creative Consilience Program through IITP grant funded by the MSIT (IITP-2025-RS-2020-II201819). We also thank the members of Computer Architecture & Systems research lab in Korea University for their useful comments and discussions, as well as the anonymous reviewers for their helpful feedback.

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

# Appendix

In this appendix, each section provides the following:

- **Section A (Technical Appendix):** detailed hardware and software configurations used in our experiments, including system specifications and common settings.
- **Section B (Parallel Training Methods):** comprehensive background on major distributed training paradigms (Data, Pipeline, and Tensor Parallelism).
- **Section C (Additional Motivation Analyses):** extended motivation analyses across different model scales, datasets, and architectures (GPT-2, ViT-B, LLaMA2-7B, CodeLLaMA-34B, and LLaMA3.2-11B-Vision), confirming (1) the feasibility of bypassing per-block MHA outputs, and (2) the pivotal role of the first MHA output across transformer variants and domains.
- **Section D (Additional Evaluation Results and Analyses):** further ablation studies, analyses on information-dilution mitigation, and inference acceleration results demonstrating how FAL improves both training and inference efficiency while preserving model stability and quality through effective reuse of the first attention output.
- **Section E (Evaluation of Generalizability to Transformer Variants):** evaluation of FAL and FAL+ on diverse transformer variants (e.g., GQA-based, MoE-based, and ViT architectures) to confirm adaptability across attention mechanisms and modalities.

## A  Technical Appendix

### A.1  Hardware & Software

We performed the experiments using PyTorch and Colossal-AI on our server and a public cloud service.

**Common Settings**

- Version of PyTorch: 2.2.2
- Version of CUDA: 12.3
- Version of Colossal-AI: 0.4.0

**System-1**

- Operating system: Ubuntu 20.04.6
- CPU: AMD EPYC 7542 32-Core
- GPU: NVIDIA RTX 3090 24GB X 4
- Interconnect: PCIe Gen4 x16 (64GB/s)

**System-2**

- Operating system: Ubuntu 20.04.6
- CPU: AMD Ryzen Threadripper 3970X 32-Core
- GPU: NVIDIA RTX 4090 24GB X 2
- Interconnect: PCIe Gen4 x16 (64GB/s)

**System-3**

- Operating system: Ubuntu 20.04.6
- CPU: AMD Ryzen Threadripper 3970X 32-Core
- GPU: NVIDIA RTX A6000 48GB X 2
- Interconnect: PCIe Gen4 x16 (64GB/s)

**System-4 (Public Cloud)**

- Operating system: CentOS 7.9
- CPU: Intel Xeon Emerald Rapids (Platinum 8558) / 2.10GHz (48-core) / 2 socket
- GPU: NVIDIA H200
- Interconnect: NVIDIA NVLink (900GB/s)

**Figure 1 (d)**

- System: 4
- GPU#: 8
- Model: GPT-2 774M
- Sequence Length: 1024
- Batchsize: 128

**Figure 3 (a), (b), Figure 4 (a), (b)** Motivation analyses are done with pretrained GPT-2 model on four different text datasets – WikiText-2 [18], PTB [19], BookCorpus [20], and CC-News [21].

- System: 1
- Model: GPT-2 117M (pretrained: openai-community/gpt2-large [59] from Hugging Face)
- Max Sequence Length: 1024

**Figure 6** We employ FlashAttention [41] and mixed-precision training [60] in all experiments to maximize tensor core utilization and overall training efficiency. We benchmark using the largest batch size (in powers of two) supported under each training setting.

- System: 4, 1
- GPU#: [2, 4, 8]
- Model: [GPT-2 774M, 1.5B, 2.5B, 8.3B]
- Batchsize: System4: 64 (774M, 2GPU), 16 (1.5B, 2GPU), 16 (2.5B, 2GPU), 8 (8.3B, 2GPU), 64 (774M, 4GPU), 32 (1.5B, 4GPU), 32 (2.5B, 4GPU), 16 (8.3B, 4GPU), 128 (774M, 8GPU), 64 (1.5B, 8GPU), 64 (2.5B, 8GPU), 32 (8.3B, 8GPU) System1: 4 (774M, 2GPU), 2 (1.5B, 2GPU), 8 (774M, 4GPU), 4 (1.5B, 4GPU)
- Sequence Length: 1024

**Figure 7**

- System: 1
- GPU#: 2
- Model: GPT-2 774M
- Total Batchsize: 32 (used gradient accumulation)
- Sequence Length: 1024
- Epochs: 1
- Learning rate: 0.0001
- Weight decay: 0.001
- Dropout: 0

**Figure 8 (a)** We compare GPT-2 and FAL under both the minimum (1) and maximum batch sizes for each GPU setting. We also evaluate the speedup with and without acceleration techniques — specifically, FlashAttention — on each GPU configuration.

- System: [1, 2, 3]
- GPU#: [1]
- Model: GPT-2 774M
- Sequence Length: 1024

**Figure 8 (b)**   We use NVIDIA Nsight Systems [61] to profile GPU performance, including SM utilization, warp occupancy, tensor core usage, and memory bandwidth.

- System: 1
- GPU#: 1
- Model: GPT-2 774M
- Sequence Length: 1024

**Table 1** We train each architecture on OpenWebText [38], an open-source replication of the WebText dataset originally used to train GPT-2. The dataset comprises approximately 41.7 GB of text, corresponding to 4 billion tokens. Given our limited computational resources, we use a compute-efficient batch size of 32, which has been shown to be sufficient for stable hyperparameter transfer in μP-based training [62, 63]. To evaluate language understanding performance, we report zero-shot results on the SuperGLUE benchmark [8], which includes BoolQ [42], CB [43], COPA [44], MultiRC [45], ReCoRD [46], RTE [47], WiC [48], and WSC [49]. No finetuning or additional training was performed on any task. CB and ReCoRD are evaluated using F1 score, while the remaining tasks use accuracy.

- System: 1
- Epochs: 1
- GPU#: 4
- Model: GPT-2 774M, 1.5B
- Parallel setting: 2TP/2DP
- Total batchsize: 32 (used gradient accumulation)
- Sequence Length: 1024
- Learning rate: 0.0001
- Weight decay: 0.001
- clip-grad-norm: 1
- embd-pdrop: 0.1

**Figure 9**   Motivated by Cramming [50], which demonstrated that scaling laws [1] can be observed even under small-scale, fast-training settings, we compare FAL and FAL+ to the standard pre-LN architecture by stacking transformer blocks with depths of 36 (equivalent to GPT-2 774M), 48 (GPT-2 1.5B), and 60. To evaluate scalability, we stack the transformer blocks of a pre-LN masked language model architecture based on BERT-Large [64].

Training settings follow the original Cramming paper, including a budget-based one-cycle learning rate scheduler [65] and batch size ramp-up for 24 hours on 1 GPU. Models of the same scale are trained under identical system configurations.

- System: 1, 2
- GPU#: 1
- Hidden size: 1024
- Intermed size: 4096
- Nonlinear: GELU
- Max sequence length: 128
- Number of transformer block: 36, 48, 60
- Final Batchsize: 8192 (used gradient accumulation)
- Learning rate: 0.0001
- Weight decay: 0.001
- Clip-grad-norm: 1
- Hidden dropout probability: 0.1
- Attention dropout probability: 0.1
- Embedding dropout probability: 0.1

# B    Parallel Training Methods

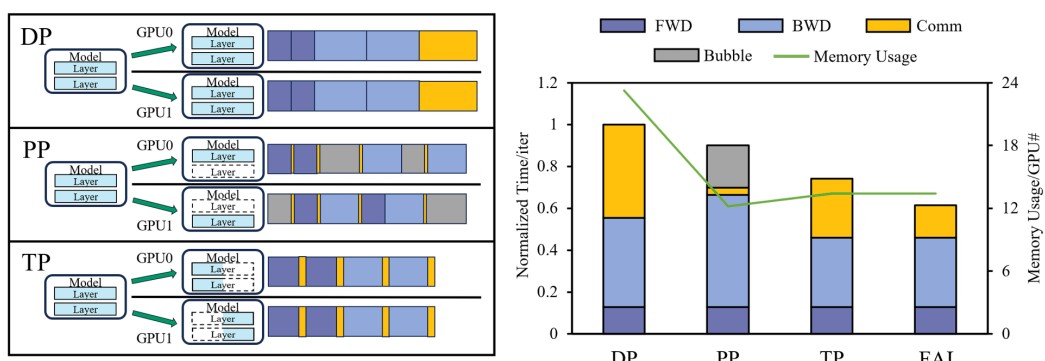

Figure 10: Comparison of Data parallelism(DP), Pipeline Parallelism(PP), Tensor Parallelism(TP). We stack GPT-2 blocks until DP can handle them (number of blocks: 42), using OpenWebText (max sequence length: 1024) for comparison on two NVIDIA RTX 3090 GPUs connected via PCIe.

To address challenges of training the large-scale models, employing multiple distributed GPUs along with parallel training methods [66, 15, 4, 67, 68, 16, 69, 70] has become a common practice. These methods encompass a range of parallelism paradigms, including Data Parallelism (DP), Pipeline Parallelism (PP), and Tensor Parallelism (TP). DP duplicates the entire model across multiple distributed GPUs. Each GPU then trains the duplicated model with different data batches and synchronizes the trained gradients for unified updates [15, 69, 71]. While DP is effective for smaller models, it results in significant memory and communication overhead for larger models as each GPU needs to retain the model duplicates and synchronize the large amount of parameters.

PP and TP have been proposed to address the scalability issue. PP [72, 16] partitions layers of a model across the GPUs. A batch is split into smaller microbatches, and training of different layers is pipelined with the microbatches across the GPUs. However, to ensure consistent weight updates for a particular batch (without being affected by the weight updates from the other batches), GPUs need to synchronize the weight updates of microbatches for every batch. This introduces pipeline bubbles where some GPUs (which process former microbatches of a batch) need to wait for the weight updates from the other GPUs (which process latter microbatches of the batch) delaying the entire training process [72]. TP [4], on the other hand, distributes matrix multiplications within each transformer layer (i.e., MHA and MLP) across the GPUs. Each GPU handles a portion of the matrix multiplications in parallel, without having model duplicates or pipeline bubbles, making it highly effective for large-scale models.

Although TP is receiving much attention recently for its scalability benefit to further enhance memory efficiency and latency with large-scale models [5, 6, 55], its further efficiency is still limited by the communication overhead. Fig. 10 illustrates the train time and memory usage comparison between DP, PP and TP. While TP shows the fastest training time among three methods as it does not require communication of full parameters and pipeline bubbles, frequent communication between GPUs is still required to process synchronized and complete intermediate activations and gradients from MHA and MLP. As a result, a large portion of the training time is devoted to these communications (37.9% of the training time), resulting in a notable decrease in training efficiency.

To further enhance the potential of TP for fast large model training, we propose FAL, which eliminates intra-block data communication by harnessing the output of the MHA in the first (i.e., bottom-most) transformer block for the MLP's inputs, instead of using the output of the MHA in the same block.

## C    Additional Motivation Analyses

### C.1    Motivation Analyses in Different Scale

Fig. 11 shows the CKA similarity scores for MHA outputs, MLP inputs (*Residual + MHA*), and MLP outputs across adjacent blocks (conducted on GPT-2 774M and 1.5B). As shown in Fig. 11, even in the case of larger models, the MLP input remains highly similar despite significantly changing MHA

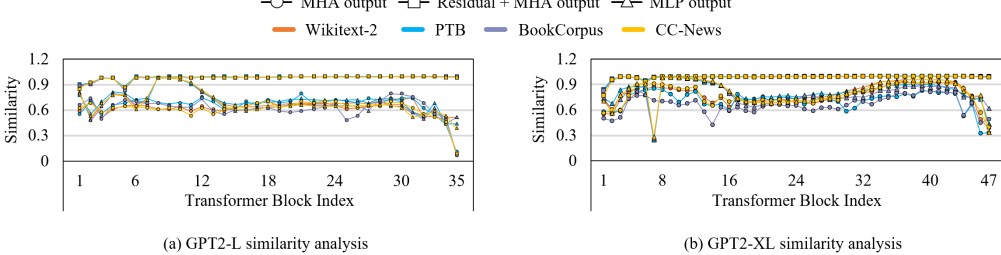

(a) GPT2-L similarity analysis

(b) GPT2-XL similarity analysis

Figure 11: CKA-based similarity analysis of GPT-2 774M and 1.5B. across successive Transformer blocks. The x-axis shows the block index, and the y-axis shows the similarity (CKA) between consecutive MHA output, Residual + MHA output (i.e., the MLP input), and the MLP input

output. This demonstrates that, regardless of the model size, MLP may not require the most recent MHA output (i.e., the output of the MHA within the same block).

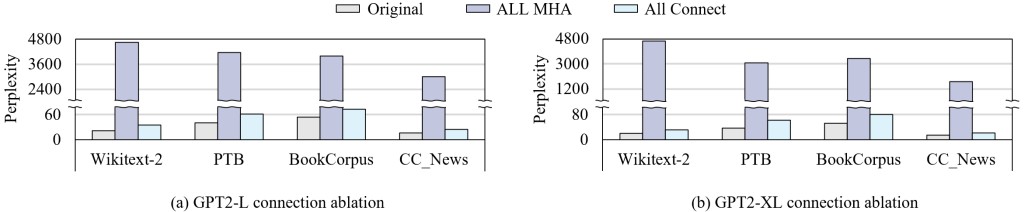

(a) GPT2-L connection ablation

(b) GPT2-XL connection ablation

Figure 12: Connection ablation results measured by perplexity with GPT-2 774M and 1.5B. "Original" denotes the unaltered model. "All MHA" removes every MHA layer. "All Connect" removes every direct MHA-MLP connection.

Fig. 12 illustrates two scenarios: removing all MHAs (*All MHA*) versus removing all MHA-MLP connections (*All Connect*), measured by perplexity on GPT-2 774M and 1.5B. As expected, removing *All MHA* severely degrades model quality. In contrast, removing *All Connect* recovers a significant portion of the lost performance compared to removing the entire MHAs, and this recovery becomes even more pronounced with larger models (though still not fully reaching the original performance). This suggests that bypassing MHA is a better option for larger models.

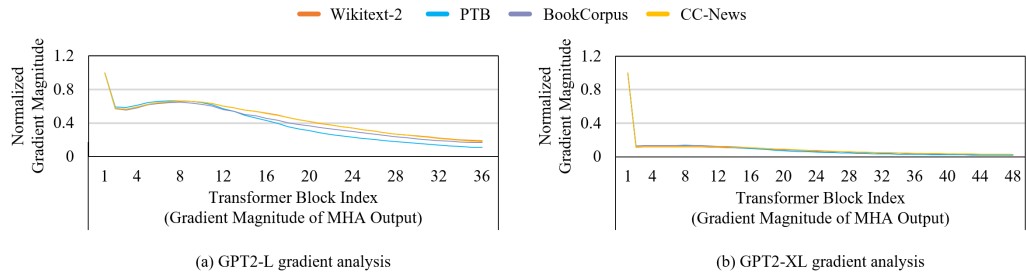

(a) GPT2-L gradient analysis

(b) GPT2-XL gradient analysis

Figure 13: Normalized gradient magnitude of the MHA outputs across Transformer blocks in GPT-2 774M and 1.5B for different datasets. the x-axis represents the block index.

Fig. 13 shows the gradient magnitude of each MHA output on larger scale (774M and 1.5B). As shown in Fig. 13, even in the case of larger models, first MHA output consistently exhibits the highest gradient magnitude. This confirms our finding that perturbations in the earliest attention result have a disproportionately large impact on final predictions, regardless of model size.

Fig. 14 shows the perplexity after omitting the MHA from individual transformer blocks with 774M and 1.5 scale GPT-2. As shown in Fig. 14, removing the first attention causes a far larger perplexity increase than removing later layers, verifying the crucial role of the first attention in language modeling. These findings align with the well-known psychological phenomenon of the primacy effect [26], commonly summarized as "first impressions matter." The primacy effect of the first

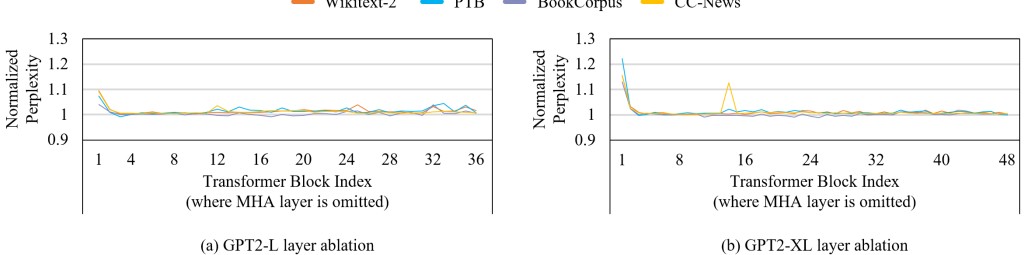

(a) GPT2-L layer ablation

(b) GPT2-XL layer ablation

Figure 14: Layer ablation results measured by perplexity with GPT-2 774M and 1.5B. the x-axis indicates the index of the transformer block from which the MHA is omitted.

attention is not limited to a specific model architecture — previous works also identified the prominent impact of the first attention layer across various attention mechanisms and tasks [27, 28, 29].

## C.2    Motivation Analyses with Different Task & Model Architecture

Beyond scaling analyses, we further validate our motivation across different tasks and model architectures, including ViT-B (86.6M), LLaMA2-7B, CodeLLaMA-34B, and LLaMA3.2-11B-Vision.

### C.2.1    ViT

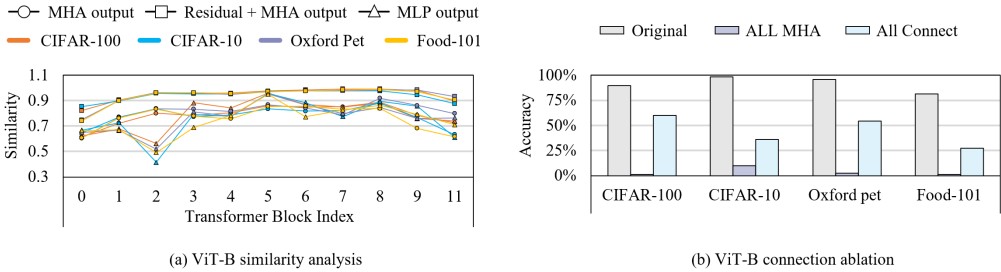

(a) ViT-B similarity analysis

(b) ViT-B connection ablation

Figure 15: (a) CKA-based similarity analysis of ViT. across successive Transformer blocks. The x-axis shows the block index, and the y-axis shows the similarity (CKA) between consecutive MHA output, Residual + MHA output (i.e., the MLP input), and the MLP input (b) Connection ablation results measured by accuracy. "Original" denotes the unaltered model. "All MHA" removes every MHA layer. "All Connect" removes every direct MHA-MLP connection.

Fig. 15 (a) shows the CKA similarity scores for MHA outputs, MLP inputs (*Residual + MHA*), and MLP outputs across adjacent blocks (conducted on ViT-B). As shown in the Figure, even in the case of vision task, the MLP input remains highly similar despite the MHA output changing significantly. This confirms our findings that MLP may not always require the most recent MHA output (i.e., the output of the MHA within the same block), regardless of the domain.

Fig. 15 (b) illustrates two scenarios: removing all MHAs (*All MHA*) versus removing all MHA-MLP connections (*All Connect*), measured by accuracy on ViT-B. As expected, removing *All MHA* severely degrades model quality. In contrast, removing *All Connect* recovers a large portion of the lost performance compared to removing the entire MHAs, however this recovery becomes smaller with smaller vision models. This suggests that simply bypassing MHA on small-scale vision models may harm their accuracy.

Fig. 16 (a) shows the normalized gradient magnitude of each MHA output in the encoder-based ViT-B [73]. Although the effect is less pronounced than in language models, the first MHA output still consistently exhibits the highest gradient magnitude.

Fig. 16 (b) shows the accuracy after omitting the MHA from individual transformer blocks with ViT-B. As shown in the Figure, removing the first attention causes a far larger accuracy drop than removing later layers (except the last), verifying the crucial role of the first attention in language modeling. The

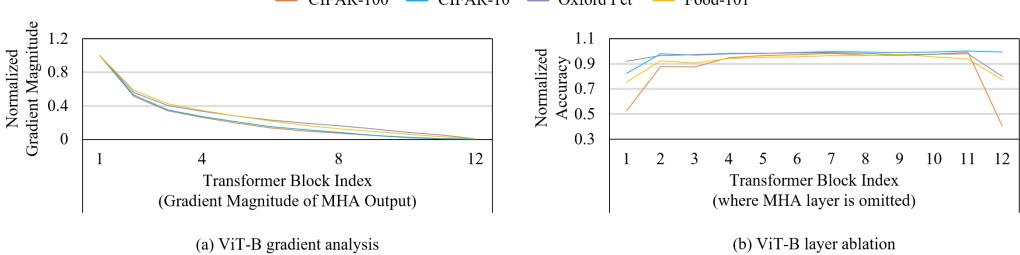

(a) ViT-B gradient analysis

(b) ViT-B layer ablation

Figure 16: (a) Normalized gradient magnitude of the MHA outputs across Transformer blocks with ViT for different datasets. The x-axis represents the block index. (b) Layer ablation results measured by accuracy with ViT. The x-axis indicates the index of the transformer block from which the MHA is omitted.

importance of the final attention likely stems from the fact that ViT uses a domain-specific classifier based on the output of the final Transformer block [73].

### C.2.2 LLaMA

To further validate the generality of our findings across modalities and scales, we extend our motivational analyses to larger models: LLaMA2-7B (language), CodeLLaMA-34B (code generation), and LLaMA3.2-11B-Vision (multilingual vision-language). We use WikiText for LLaMA2-7B, The Stack dataset [74] for CodeLLaMA-34B, and the COCO captioning dataset for LLaMA3.2-11B-Vision.

Table 3: Similarity Analysis Results (Metric: CKA Similarity ± std)

| Activation | LLaMA2-7B | CodeLLaMA-34B | LLaMA3.2-11B-Vision |
|---|---|---|---|
| **Attn Out** | 0.60 ±0.14 | 0.66 ±0.13 | 0.84 ±0.10 |
| **MLP In** | 0.98 ±0.03 | 0.99 ±0.03 | 0.98 ±0.02 |
| **MLP Out** | 0.70 ±0.23 | 0.80 ±0.18 | 0.80 ±0.20 |

Table 3 reports the similarity analysis results, measured by CKA similarity (mean ± std). Across all three models, we observe that the MLP inputs remain highly similar to each other (LLaMA2-7B: 0.98, CodeLLaMA-34B: 0.99, LLaMA3.2-11B-Vision: 0.98), whereas the attention/MLP outputs vary more significantly (0.60–0.84). This analysis shows that the residual path already accumulates sufficient attention signals, making the MLP input less sensitive to the most recent MHA output. The same trend holds even for larger-scale (7B–34B) and multimodal (vision–language) models, extending our earlier motivation analyses 3.1 and supporting the validity of reconfiguring MHA–MLP connections in FAL.

Table 4: Layer Ablation vs. Connection Ablation (Metric: Validation Perplexity)

| Setting | LLaMA2-7B | CodeLLaMA-34B | LLaMA3.2-11B-Vision |
|---|---|---|---|
| **Original** | 7.39 | 1.80 | 25.12 |
| **Remove Layer** | 5339.99 | 1484.52 | 3995.84 |
| **Remove Connection** | 892.12 | 37.06 | 504.72 |

Table 4 reports validation perplexity under two ablations: (1) removing the entire MHA layer, and (2) removing only the direct MHA–MLP connection. Across all three models, removing entire layers leads to catastrophic degradation (e.g., PPL >1000), while removing only the connections recovers a substantial portion of performance. Although the connection–removed models still fall short of the original baseline, they consistently perform far better than the layer–removed ones, and the recovery effect becomes more pronounced at larger scales. These results show that reconfiguring MHA–MLP connections causes far smaller degradation than removing entire layers, yet still falls short of fully matching the original performance. A stable alternative signal is needed, which is exactly what FAL provides by reusing the impactful first attention.

Table 5: Gradient Analysis (Metric: Gradient L1 Norm)

| Block | LLaMA2-7B | CodeLLaMA-34B |
|---|---|---|
| **1st** | 505.99 | 321.65 |
| **2-End avg ±std** | 85.90 ±102.00 | 46.24 ±60.39 |
| **Ratio (1st/avg)** | 5.9× | 7.0× |

Table 5 reports the gradient analysis of MHA outputs in LLaMA2-7B and CodeLLaMA-34B. The first attention block shows a much larger gradient magnitude (505.99 and 321.65) compared to the average of later blocks (85.90 and 46.24, respectively). On average, the first block gradients are 5.9× and 7.0× larger than subsequent ones. These results highlight that the first attention exerts a disproportionately strong influence on final predictions. Thus, reusing this impactful signal in FAL remains well-justified across larger models and diverse tasks.

Table 6: Layer Ablation per Block (Metric: Validation Perplexity)

| Block | LLaMA2-7B | CodeLLaMA-34B | LLaMA3.2-11B-Vision |
|---|---|---|---|
| **Original** | 7.39 | 1.80 | 25.12 |
| **1st** | 34.37 | 4.56 | 40.94 |
| **2-End avg ±std** | 4.37 ±1.37 | 1.81 ±0.01 | 24.75 ±1.25 |
| **Ratio (1st/avg)** | 7.9× | 2.5× | 1.7× |

Table 6 reports the effect of ablating individual attention layers in LLaMA2-7B, CodeLLaMA-34B, and LLaMA3.2-11B-Vision. Removing the first attention block causes a far larger degradation in validation perplexity (e.g., 34.37 vs. 7.39 for LLaMA2-7B, 4.56 vs. 1.80 for CodeLLaMA-34B, and 40.94 vs. 25.12 for LLaMA3.2-11B-Vision) compared to ablating later layers, whose impact remains relatively small. The impact of removing the first attention is consistently larger than that of later layers, up to 7.9× in LLaMA2-7B. These results show that the first attention is disproportionately important across scales and modalities. Its removal uniquely destabilizes the model, whereas later attentions contribute far less. This further supports that reusing the first attention in FAL remains well-justified across larger and more diverse models.

## D   Additional Evaluation Results and Analyses

### D.1   Ablation study

We conduct two ablations to verify the effectiveness of reusing the first attention output in FAL and FAL+. Table 7 shows the validation perplexity and training time.

Table 7: Comparison of validation perplexity and training time using Openwebtext dataset (GPT-2 774M)

| Model | Perplexity | Training time |
|---|---|---|
| GPT-2 774M (Baseline) | 17.75 | 13.2 days |
| **FAL** | **17.55** | 8.6 days |
| **FAL+** | **17.24** | 13.2 days |
| Ablation1 | 21.34 | 13.2 days |
| Ablation2 | 17.98 | 8.6 days |

**Ablation1 (leveraging latest attention).**   Simple addition of normalized outputs within a block degrades the quality, calling for the necessity of the first attention signal. We apply the same LN + LN structure from FAL using the latest attention output instead of the first attention.

$$X_i + \mathrm{MHA}_i(\mathrm{LN}(X_i)) + \mathrm{MLP}_i(\mathrm{LN}(X_i) + \mathrm{LN}(\mathrm{MHA}_i(\mathrm{LN}(X_i)))) \qquad (3)$$

This leads to a significantly higher perplexity (21.34) compared to the baseline (17.75), suggesting that the latest attention output does not provide a stable or beneficial signal for MLP inputs under this reconfiguration.

**Ablation2 (removing connection without first).** Retaining only the first MHA-MLP connection is not sufficient to retain the first attention signal. We remove all MHA-MLP connections except for the first one.

$$\begin{cases} X_1 + \text{MHA}_1(\text{LN}(X_1)) + \text{MLP}_1\big(\text{LN}(X_1) + \text{MHA}_1(\text{LN}(X_1)))\big), & \text{if } i = 1, \\ X_i + \text{MHA}_i(\text{LN}(X_i)) + \text{MLP}_i\big(\text{LN}(X_i)\big), & \text{otherwise.} \end{cases} \tag{4}$$

Although perplexity (17.98) remains comparable to the baseline, it still degrades model quality compared to FAL. This implies that merely connecting the first attention once is not sufficient to maintain the overall performance.

**Comparison to FAL and FAL+.** FAL reuses the first MHA output in each block. This reuse, aided by an additional LN to balance the first and residual signals, improves perplexity (17.55) and reduces training time (8.6 days). FAL+ augments the original connections with the first attention, achieving an even lower perplexity (17.24) but at a training time similar to GPT-2. These results confirm that properly integrating the first attention output is crucial for both efficiency and model quality.

**GPT-2.**
$$X_i + \text{MHA}_i(\text{LN}(X_i)) + \text{MLP}_i(\text{LN}(X_i + \text{MHA}_i(\text{LN}(X_i)))) \tag{5}$$

**FAL.**
$$X_i + \text{MHA}_i(\text{LN}(X_i)) + \text{MLP}_i(\text{LN}(X_i) + \text{LN}(\text{MHA}_1(\text{LN}(X_1)))) \tag{6}$$

**FAL+.**
$$X_{i+1} = \begin{cases} X_1 + \text{MHA}_1(\text{LN}(X_1)) \\ \quad + \text{MLP}_1\big(\text{LN}(X_1) + \text{MHA}_1(\text{LN}(X_1)))\big), & \text{if } i = 1, \\ X_i + \text{MHA}_i(\text{LN}(X_i)) \\ \quad + \text{MLP}_i\big(\text{LN}(X_i + \text{MHA}_i(\text{LN}(X_i))) + \text{LN}(\text{MHA}_1(\text{LN}(X_1)))\big), & \text{otherwise.} \end{cases} \tag{7}$$

**Ablation with Other Layers.** To further verify the benefit of reusing the impactful first attention, we compare FAL+ with variants that reuse the output of other MHA layers (2nd, 3rd, and so on). We adopt FAL+ in a 48-block configuration (see Fig. 9) and train for 500k steps under the same one-cycle schedule with a batch-size ramp-up to 8,192, ingesting 1.02B tokens regardless of hardware speed.

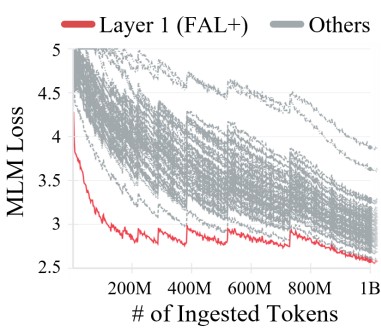

Fig. 17 shows the MLM loss comparison of FAL+, which reuses the first attention, against variants that reuse later-layer attentions. As shown in Fig. 17, reusing later-layer attentions consistently underperforms compared to the first attention. This implies that reusing weaker or less dominant signals is not as effective as leveraging the impactful first attention, confirming that the first attention provides a uniquely stable and beneficial feature for reconfiguration.

Figure 17: Loss comparison using other MHA's output

## D.2 How FAL Mitigates Information Dilution in Deep Transformers

In a standard Pre-LN Transformer, each block processes only its immediate predecessor's output, analogous to unrolling an RNN over depth. As depth grows, early-layer signals must traverse many transformations and risk dilution or loss—much like long RNN sequences forget initial states. FAL breaks pure sequential dependence by feeding the first-layer attention output directly into every later block, akin to self-attention's ability to attend globally. Concretely, we normalize the first MHA output once and then add it to each block's input: $MLP_{input_i} = LN(X_i) + LN(MHA_1(LN(X_1)))$

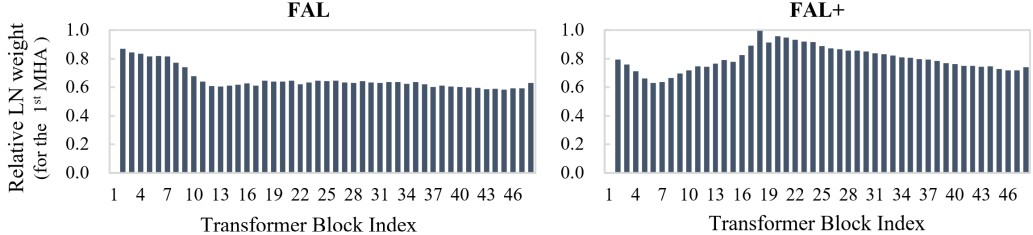

Figure 18: Relative LN scaling parameters connected to the first MHA output, normalized to average LN scaling across layers.

Residual connections alone cannot prevent the first signal from fading: activation variance accumulates layer by layer [9], eventually washing out early-layer information — much like how self-attention's $O(n^2)$ interactions dilute key dependencies over very long sequences [75]. FAL sidesteps this pitfall by reusing normalized first-attention tensor at each block, reinforcing the most salient initial context without extra overhead — echoing cognitive insights that rethinking a first impression can lead to better decisions in deep reasoning [30].

Figure 18 shows the average LN scaling parameters ($\gamma$) for the terms connected to the first MHA output, normalized by the average LN scaling across layers. Across depth, both FAL and FAL+ consistently assign non-negligible weights (roughly equivalent to a 0.58:1–1:1 ratio compared to the current block input), indicating that later blocks actively and adaptively incorporate the first-attention signal after training. This dynamic weighting suggests that FAL can alleviate information dilution by adaptively reinforcing the initial signal across depth.

## D.3    Inference Acceleration of FAL

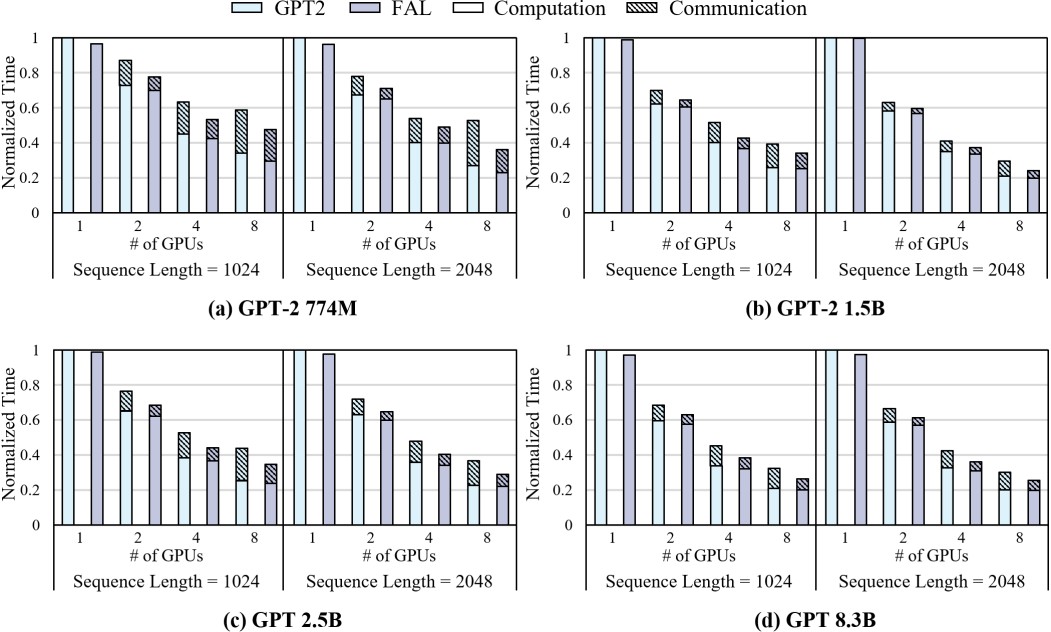

Figure 19: Normalized Multi-GPU Inference Time of GPT-2 and FAL.

TP is often used to accelerate the inference execution, as it can accelerate per-request latency unlike other two parallelism methods (i.e., Data Parallelism and Pipelined Parallelism) [7, 76]. In order to evaluate the inference acceleration that can be further achieved by FAL in TP, we measure the forward step time without gradient calculation which is aligned with the Time To First Token (TTFT) in the inference execution.

Fig. 19 shows the normalized forward step time of GPT-2 and FAL on a multi-GPU server with NVLink (System 4), across model sizes ranging from 774M to 8.3B and sequence lengths of 1024 and 2048. As shown in the figure, tensor parallelism (TP) reduces per-request inference time by utilizing multiple GPUs. With 8 GPUs, TP achieves an average speedup of 56.5% for a sequence length of 1024 (up to 67.6%), and 62.8% for a sequence length of 2048 (up to 70.5%). However, as the number of GPUs increases, the degree of acceleration is limited by the communication overhead between GPUs. In such cases, FAL improves inference performance over GPT-2 by (1) significantly reducing inter-GPU communication and (2) increasing intra-GPU parallelism. Across configurations from 1 to 8 GPUs, FAL reduces inference time by 11.1% on average (up to 31.6%).

# E   Evaluation of Generalizability to Transformer Variants

## E.1   Loss comparison using variants of Multi Head Attention

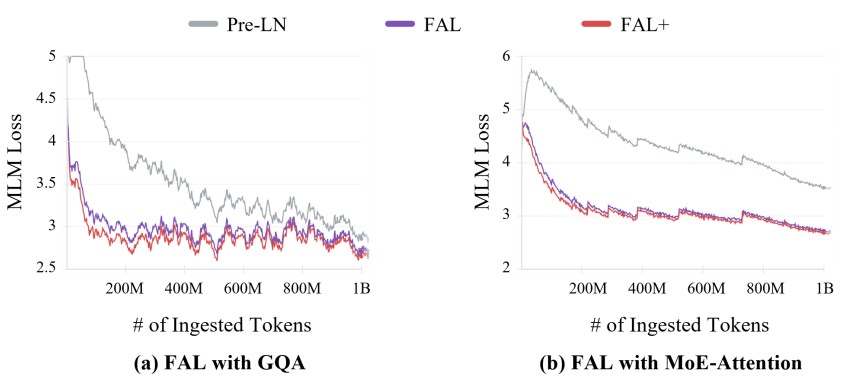

(a) FAL with GQA

(b) FAL with MoE-Attention

Figure 20: Loss comparison across different attention mechanisms: Grouped Query Attention (GQA), and MoE-based Attention (MoE-Attention).

FAL and FAL+ can be applied to various Pre-LN based transformer variants, such as LLaMa (with Grouped Query Attention (GQA)), and Switch Transformer (with Mixture of Experts (MoE)) to improve the efficiency and model quality. To evaluate the generalizability of FAL and FAL+ to these variants, we measure token ingestion efficiency before and after applying FAL and FAL+ to GQA-based [77] and MoE-based attention models [58]. We adopt FAL and FAL+ in a 48-block configuration (see Fig. 9). To ensure consistent scheduling across hardware setups, we train for 500,000 steps using a step-based one-cycle learning rate scheduler. The batch size is ramped up to a maximum of 8,192 by step 300,000, resulting in a total of 1.02B tokens ingested regardless of hardware speed.

Fig. 20 (a) shows the comparison of FAL and FAL+ when applied to GQA. Each attention layer uses GQA with two groups. This setup differs from standard Multi Head Attention (MHA) primarily in the key/value projections, and becomes equivalent to MHA when the number of groups equals the number of heads. The results closely resemble those observed with standard MHA. The loss gap between the proposed architectures and the baseline remains consistent, demonstrating that the gains from FAL and FAL+ extend robustly to this efficient attention variant.

Fig. 20 (b) shows the comparison of FAL and FAL+ when applied to MoE-based Attention. We follow the configuration of MoE-based Attention introduced in the Switch Transformer, which was found to be unstable and thus not included in the final architecture. Each expert in the MoE attention has its own query projection and tied key/value projections. One of two experts is activated per attention layer. Unlike the instability observed when using Switch layers in attention, FAL and FAL+ do not suffer from gradient instability. The loss gap between the proposed architectures and the baseline remains consistent, demonstrating that the gains from FAL and FAL+ extend robustly to sparsely activated MoE attention mechanisms.

Table 8: Comparison of validation accuracy using ImageNet dataset (ViT-B 86.6M)

| Dataset | ViT (Baseline) | FAL | FAL+ |
|---------|---------------|------|------|
| ImageNet | 79.06% | 78.76% | **79.20**% |

## E.2   Accuracy comparison using other kinds of Task

We also evaluate FAL and FAL+ on the Vision Transformer (ViT-B 86.6M) architecture using the ImageNet dataset. As shown in Table 8, FAL slightly reduces accuracy compared to the baseline (79.06% vs. 78.76%), whereas FAL+ achieves a higher accuracy (79.20%). This result indicates that reusing the first attention output can also benefit vision tasks, particularly when combined with the original attention connections as in FAL+.

