# OpenReview forum: "First Attentions Last: Better Exploiting First Attentions for Efficient Parallel Training"
_NeurIPS.cc/2025/Conference — NeurIPS 2025 poster_

### Official Review · Reviewer_R6Kr · 2025-07-04

**Clarity:** 3
**Significance:** 3
**Originality:** 2
**Rating:** 4
**Confidence:** 3

**Summary:**

This paper addresses the computation overhead in large language model pretraining, particularly within the context of tensor parallelism (TP). The authors noticed that the all-reduce operation in the connection of multi-head attention (MHA) and feed-forward layer (MLP) forms a bottleneck. Based on two empirical findings: 1) MHA-MLP connections can be reconfigured and 2) first-layer attention output is more important, the authors propose two new architectures FAL and FAL+, which add further skip-connection for each MLP layer with the first-layer attention outputs. Experiments with GPT-2 show that the proposed method is more compute efficient and could achieve better performance than the baseline.

**Questions:**

1. In large-scale modeling, the residual connection is unlikely to be a major bottleneck, as major computation is spent on the attention and MLP layers. It would be significant to show timing analysis with model scaling.
2. As FAL/FAL+ requires storing the first MHA output, they may incur further memory overhead. Could you provide a detailed analysis from the memory perspective? This could be significant as sequence length, batch size and model scale increases.
3. In Table 1, at 1.5B, FAL+ achieves better PPL but worse SuperGLUE. Could you provide some potential reasons for this?

**Ethical Concerns:**

["NO or VERY MINOR ethics concerns only"]

**Limitations:**

The observation, like 44% training time reduction, is likely not generalizable to real larger models.

**Paper Formatting Concerns:**

I didn't see major issues.

**Quality:**

3

**Strengths And Weaknesses:**

Strengths:

- the analysis is intuitive and the method is easy to understand and implement
- FAL and FAL+ achieve decent performance on several evals, either improved efficiency and/or better quality, compared to the baseline


Weaknesses:
- GPT-2 is a relatively old architecture. Findings may not generalize well to modern architectures, like LLaMa and Gemma.
- Model scale is generally too small, under which the conclusion is unlikely to generalize to larger setups.

---

> ### Author Rebuttal · Authors · 2025-07-31
>
> We appreciate your valuable feedback.
>
> ### **W1:** Generality Beyond GPT-2
> To demonstrate FAL’s broad applicability, we include the following additional evaluations:
>
> **Transformer Variants (GQA & MoE): Appendix E.1** shows that applying FAL/FAL+ to Grouped Query Attention —as used in LLaMA and Gemma— and to MoE-based attention yields loss-reduction curves and stable quality gains nearly identical to those on standard MHA, confirming that our “first-attention matters” insight extends to these modern variants. **Appendix E.2** reports ImageNet accuracy for ViT-B (86 M) where FAL+ actually **improves** top-1 accuracy (79.20% vs. 79.06%) and FAL matches baseline, indicating cross-domain robustness.
>
> **Visaon-Language and Code Models:** To further validate cross-model applicability, we have extended our **motivation analysis to:**
> **Llama2-7B** (7 B parameters)
> **CodeLlama-34B** (34 B parameters)
> **VisionLlama-11B** (11 B parameters, multilingual vision-language)
>
> The experimental results confirm (1) the feasibility of bypassing per-block MHA outputs, and (2) the pivotal role of the first MHA output across architectures and domains.
> For the detailed results, please see our response to **W1&Q2** of Reviewer **eLP5**.
>
> **Supporting Literature:** Prior works such as (Liu, Liyuan, et al., 2020, Liu, Zichang, et al., 2023) report minimal activation changes across layers, and (Chen, et al., 2024, Zhang, et al., 2024) the important role of first MHA which are consistent with our findings. These extensions firmly establish that FAL’s principles—exploit the first MHA output to reduce overhead by replacing subsequent direct MHA–MLP connections— hold across modern architectures, like LLaMa and Gemma.
>
> - Liu, Liyuan, et al. "Understanding the difficulty of training transformers." arXiv preprint arXiv:2004.08249 (2020).
> - Liu, Zichang, et al. "Deja vu: Contextual sparsity for efficient llms at inference time." International Conference on Machine Learning. PMLR, 2023.
> - Chen, Xingwu, Lei Zhao, and Difan Zou. "How transformers utilize multi-head attention in in-context learning? a case study on sparse linear regression." Advances in Neural Information Processing Systems 37 (2024): 119573-119613.
> - Zhang, Yang, Yanfei Dong, and Kenji Kawaguchi. "Investigating layer importance in large language models." arXiv preprint arXiv:2409.14381 (2024).
>
> ### **W2&Limitations:** Model Scale
>
> While our main experiments are done on a relatively small scale compared to SOTA LLMs, both **efficiency gains and quality preservation/improvement are scalable to larger models**.
>
> **1. Performance Scalability — Model-Independent Gains from Communication Reduction**
>
>
> By cutting per-block communication in half, FAL can achieve on the order of **44 % (or greater) end-to-end training time reduction** irrespective of model size, and its gains are expected to become more pronounced for larger scales.
>
> FAL’s key benefit comes from halving the frequent all-reduce operations between each block’s MHA and MLP — precisely the communication bottleneck in Tensor Parallel training. Such communication overhead is expected to be more significant in large-scale setups, as communication bandwidth has not kept pace with compute performance; over the last two decades, raw FLOP/s per chip grew by roughly **60 000×**, whereas interconnect bandwidth improved by only **30×** (Gholami et al., 2024). This growing gap already manifests in practice. For instance, **training a 200 B-parameter model on TPU v4 Pods** shows that **communication alone accounts for 42% of training time** (Wang et al., 2022). On typical GPU-based systems — where interconnect bandwidth is **up to 10× slower** than TPU v4 (Selvan & Kanwar, 2021) — this fraction is expected to be even higher. FAL’s architecture is designed precisely for such scenarios, making it especially valuable in **large-scale GPU clusters** with slower interconnects, where communication overhead dominates. As such, FAL’s performance improvements remain robust across scales and are likely to **amplify on future systems** as the compute/communication imbalance continues to widen.
>
> **2. Model Quality — Scaling trends show greater benefit at larger depths:**
>
>
> Quality benefits of FAL and FAL+ become more pronounced for large-scale, deeper models.
>
>
> **Deep-Model Scaling Experiments:** When we increase the model depth in fast-training scaling tests (36 → 48 → 60 layers as in Fig. 9), both FAL and FAL+ consistently reduce masked-language-model (MLM) loss more rapidly than the Pre-LN baseline. While all models converge to similar losses at 36 layers, FAL and FAL+ achieve progressively lower loss as the model depth increases.
>
>
> **Rationale behind the Deep-Model Scaling Results:** In standard Pre-LN architectures, activation variance accumulates across layers, gradually washing out early-layer information (Xiong, Ruibin, et al., 2020) — **residual connections alone cannot fully preserve** the initial signal. FAL and FAL+ counteract this attenuation by repeatedly reintroducing the high-impact first MHA output at every block, **allowing each block to adaptively re‑weight and reinterpret this signal** throughout training.
>
> **Key Motivation Generality to Large-Scale Models:** When we extended our motivation analysis to larger models, **VisionLlama-11B** and **CodeLlama-34B**, we observed the same two key phenomena: **feasibility of bypassing** per-block MHA outputs without catastrophic quality loss and **pivotal role of the first MHA output** as the critical bridging signal for performance recovery. For the detailed analysis results, please see our response to **W1**.
>
> **Together**, as model depth—and thus the “distance” between input and final output—increases, repeatedly leveraging the first attention signal yields ever greater quality gains. This makes FAL and FAL+ especially valuable for state-of-the-art, large-scale transformer models.
>
> - Gholami, Amir, et al. "Ai and memory wall." IEEE Micro 44.3 (2024): 33-39.
> - Wang, Shibo, et al. "Overlap communication with dependent computation via decomposition in large deep learning models." Proceedings of the 28th ACM International Conference on Architectural Support for Programming Languages and Operating Systems, Volume 1. 2022.
> - Selvan, Aarush, and Pankaj Kanwar. "Google showcases Cloud TPU v4 Pods for large model training." Google Cloud Blog (2021).
> - Xiong, Ruibin, et al. "On layer normalization in the transformer architecture." International conference on machine learning. PMLR, 2020.
> ### **Q1:** Timing Analysis & Bottleneck Attribution
>
> Although the majority of GPU time is devoted to attention and MLP compute, multi-GPU data-movement—the all-reduce between each block’s MHA and MLP—still accounts for **25.4–31.7%** of total training time across scales (GPT-2 774M: 31.79%; 1.5B: 25.40%; 2.5B: 31.35%; 8.3B: 31.65%, on H200/NVLink).
>
> Figures 6 and 7 break down end-to-end time into:
> **Compute** (MHA + MLP forward/backward)
> **Communication** (all-reduce)
>
> FAL’s connection reconfiguration targets **multi-GPU data-movement** rather than residual-path compute. By halving the number of per-block all-reduce calls, FAL directly cuts the communication fraction, yielding up to **44% end-to-end training time reduction** (Fig. 6).
>
>
> ### **Q2:** Memory Overhead of Reusing First Attention
>
>
> FAL does not incur any additional memory overhead during training --- it rather slightly reduces total memory usage compared to the Pre-LN Transformers.
>
>
> **No Extra Activation Buffers:** To prevent storing additional activations—which would be required if the original MHA output were transformed (e.g., via multiplication, addition, or compression)—we **reposition** the first layer’s Layer Normalization (from MLP’s input to MHA’s output). This enables later blocks to reuse the output of the first layer's Layer Normalization without any recomputation as well as memory overhead. Note, every MHA output (including the first layer’s) is already cached during the forward pass to enable gradient computation.
>
>
> **Fewer LayerNorms → Slightly Reduced Memory:** FAL **reduces** total memory usage by approximately 1% compared to the baseline, primarily by reducing the number of LayerNorm operations and associated intermediate activations. Instead of applying a fresh LayerNorm in every block, FAL normalizes the first-attention output only once (in block 1) and reuses the result across all blocks.
>
>
> Unlike FAL, which is designed to minimize the communication cost without incurring additional memory overhead, **FAL+ is designed primarily to enhance model quality** by injecting additional first attention signals in each block.This augmentation introduces a small memory overhead of approximately
>
>
> ((1 × B × L × H) for activation + (2 × H) for affine parameters) × (#Transformer blocks − 1)
>
>
> where B is the batch size, L the sequence length, and H the hidden dimension — resulting in about a 1% increase in memory usage.
>
> ### **Q3:** Discrepancy Between PPL and SuperGLUE at 1.5B
> We attribute this discrepancy to a domain mismatch between the pretraining corpus (OpenWebText) and the downstream SuperGLUE benchmark. While perplexity measures language modeling quality on the pretraining distribution, SuperGLUE evaluates generalization to a diverse set of reasoning and classification tasks. Prior work (Tay, Yi, et al., 2021) has shown that improvements in perplexity do not always translate to downstream gains in the presence of domain gaps.
>
> Nevertheless, FAL+ achieves better SuperGLUE scores than GPT-2 at both 774M and 1.5B, despite slightly underperforming FAL in the 1.5B case. One possible explanation is that FAL's more constrained signal path acts as a form of implicit regularization.
> We believe that continued pretraining on more diverse and downstream-aligned corpora can further close this performance gap.
>
> - Gururangan, Suchin, et al. "Don't stop pretraining: Adapt language models to domains and tasks." arXiv preprint arXiv:2004.10964 (2020).

---

> > ### Author Response · Authors · 2025-08-05
> > **We’d appreciate your feedback on our response**
> >
> > **Dear Reviewer R6Kr,**
> >
> > Thank you for your review. We understand your concerns were about:
> > 1. generality beyond GPT-2,
> > 2. limited model scale,
> > 3. communication vs. compute bottlenecks,
> > 4. memory overhead, and
> > 5. perplexity vs. SuperGLUE discrepancy.
> >
> > To address them:
> >
> > 1. Our analyses extend to **larger models** (LLaMA2-7B, CodeLLaMA-34B) and **vision/multimodal tasks** (ViT-B, VisionLLaMA-11B). FAL also works effectively with **attention variants** (GQA, MoE). *(With apologies, due to space constraints, we’ve included detailed results in our reply to Reviewer eLP5, who raised a similar concern.)*
> > 2. FAL’s training time reduction mainly comes from **halving communication time - a bottleneck that persists even at 200B scale**. We believe **deeper models benefit more in quality** from FAL, as shown in Fig. 9.
> > 3. Even with high-speed interconnects (e.g., H200 w/ NVLink), **all-reduce still accounts for 25–32% of training time**. **FAL directly reduces multi-GPU data-movement** by cutting per-block communication.
> > 4. FAL slightly **reduces memory usage**, while FAL+ adds only **~1% predictable overhead**.
> > 5. The SuperGLUE drop at 1.5B likely reflects a **domain gap** between pretraining and evaluation tasks. Still, **FAL+ outperforms GPT-2** on SuperGLUE across scales.
> >
> > We hope these updates help clarify the scope and impact of our work, and we’d appreciate any final thoughts. We’d also be grateful if you could consider increasing the score if you feel the concerns have been sufficiently addressed.
> >
> > **Best regards,**
> >
> > The Authors

---

### Official Review · Reviewer_EX1R · 2025-07-06

**Clarity:** 3
**Significance:** 2
**Originality:** 2
**Rating:** 3
**Confidence:** 2

**Summary:**

To alleviate the communication overhead in TP training, this paper proposes FAL, a transformer architecture that selectively skips MHA for saving allreduce communications in TP, and FAL+, a variant that sends the first MHA output to intermediate layers through residual connections. The evaluations shows training speed improvement due to saved communication on GPT-2 model.

**Questions:**

1. Fig. 3(b) : was it trained from scratch, or fined tuned on a trained model?
2. If the first MHA output is important, how about layer #2, #3, etc, and combinations of them? On top of Figure 4, it may be worthwhile to provide an ablation study for FAL that uses more than just layer 1.

**Ethical Concerns:**

["NO or VERY MINOR ethics concerns only"]

**Final Justification:**

The authors have address my concerns and I would like to maintain my score.

**Limitations:**

Yes

**Quality:**

2

**Strengths And Weaknesses:**

Strength:
* This paper identifies a important problem with tensor paralleism -- frequent collective communication -- and proposes a novel solution that selectively skips intermediate layers.
* The evaluation on GPT-2 shows promising results on performance improvement, especially on cases where slow interconnect dominate training.
* The paper is well-written and easy to follow.

Weakness:
* The key motivation for the MHA-skipping design is based on GPT-2 only, and it is not clearly justified whether the observation that "MHA-MLP connections can be selectively skipped" stands on other models like LLama or Qwen.
* Some claims are flawed. For instance, line 256 states FlashAttention increases computational intensity and gives FAL more oppurtunity for overlapping compute-bound and memory-bound ops. This is not very precise as MLP is compute bound too, and this actually reduces the chance for overlapping. And FAL does not actually *overlap* them unless it can fuse the the (skipped) parallel transformer layers.

---

> ### Author Rebuttal · Authors · 2025-07-31
>
> We appreciate your valuable feedback.
>
> ### **W1:** Motivation Generality Beyond GPT-2
> Our core motivation—similarity after residual addition and the dominant role of the first attention—applies broadly to **Pre-LN-based Transformer architectures**, which underlie most modern large-scale models.
> **Vision Transformer (ViT-B)**: In Appendix C.2, we analyze the normalized gradient magnitudes on ViT-B and observe that the first MHA output remains the main contributor and the MLP input remains highly similar despite the MHA output changing significantly even in the case 518 of vision task, mirroring our GPT-2 findings.
> **Attention Variants (GQA & MoE)**: Appendix E.1 reports fast-training loss comparisons for Grouped Query Attention and MoE-based attention in a 48-block setup; FAL and FAL+ maintain consistent loss improvements over the baseline, confirming adaptability to these modern variants.
>
> To further validate cross-model applicability, we have extended our **motivation analysis to**:
>
>
> **Llama2-7B** (7 B parameters)
>
>
> **CodeLlama-34B** (34 B parameters, Code Generation)
>
>
> **VisionLlama-11B** (11 B parameters, multilingual vision-language)
>
> >Similarity Analysis Results (CKA Similarity ± std)
> >
>
> | Activation | LLaMA2-7B | CodeLLaMA-34B | VisionLLaMA-11B |
> |------------|----------|---------------|-----------------|
> | **Attn Out** | 0.60 ±0.14 | 0.66 ±0.13 | 0.84 ±0.10 |
> | **MLP In**   | 0.98 ±0.03 | 0.99 ±0.03 | 0.98 ±0.02 |
> | **MLP Out**  | 0.70 ±0.23 | 0.80 ±0.18 | 0.80 ±0.20 |
>
> >Layer Ablation vs Connection Ablation (Validation PPL)
> >
>
> | Setting | LLaMA2-7B | CodeLLaMA-34B | VisionLLaMA-11B |
> |---------|----------|---------------|-----------------|
> | **Original** | 7.39 | 1.80 | 25.12 |
> | **Remove Layer** | 5339.99 | 1484.52 | 3995.84 |
> | **Remove Conn.** | 892.12 | 37.06 | 504.72 |
>
> >Gradient Analysis (Gradient L1 Norm)
> >
>
> | Block | LLaMA2-7B | CodeLLaMA-34B |
> |-------|---------:|--------------:|
> | **1st** | 505.99 | 321.65 |
> | **2~End avg ±std** | 85.90 ±102.00 | 46.24 ±60.39 |
> | **Ratio (1st/avg)** | 5.9× | 7.0× |
>
>
>
>
> >Layer Ablation per Block (Validation PPL)
> >
>
> | Block            | LLaMA2-7B | CodeLLaMA-34B | VisionLLaMA-11B |
> |------------------|---------:|--------------:|----------------:|
> | **1st**          | 34.37 | 4.56 | 40.94 |
> | **2~End avg ±std** | 4.37 ±1.37 | 1.81 ±0.01 | 24.75 ±1.25 |
> | **Ratio (1st/avg)** | 7.9× | 2.5× | 1.7× |
>
>
>
> These experiments confirm (1) the feasibility of bypassing per-block MHA outputs, and (2) the pivotal role of the first MHA output across architectures and domains.
>
>
> ### **W2:** Overlap of Compute‐and Memory‐Bound Operations
> FAL executes the MHA and MLP modules on separate CUDA streams, allowing their operations to proceed concurrently. This allows the warp scheduler to hide latency effectively—when one warp stalls (e.g., on memory), another ready warp from a different stream can be scheduled without delay. We further clarify the interaction between MLP and attention phases as follow:
>
> **1. MLP GEMM still incurs memory stalls.**
>
> Although MLP’s large matrix multiplies are compute-heavy, each GEMM begins and ends with global‐memory loads and stores (Global Memory → Shared Memory → Registers → Cuda/Tensor Cores → Shared Memory → Cuda cores → Global Memory). These boundary memory transactions introduce unavoidable stalls, even if the core GEMM is compute-bound.
>
>
> **2. FlashAttention creates “slack” for warp‐scheduling.**
>
> FlashAttention’s higher arithmetic intensity in the attention block lengthens its compute phase relative to its memory phase. Thus, by extending attention’s compute window, FlashAttention generates more opportunities to hide MLP’s boundary stalls behind attention computation.
>
> **3. Empirical GPU‐utilization gains.**
>
> Figure 8-(b) shows that FAL with FlashAttention achieves higher sustained utilization of SMs and memory pipelines compared to the Pre-LN baseline, confirming that attention’s heavier compute helps to hide MLP memory stalls.
>
>
> **4. Stronger effect in tensor-parallel settings.**
>
> In multi-GPU tensor-parallel training, MLP GEMMs are split across ranks, shortening the compute window of MLP. FAL’s attention-compute overlap mechanism becomes even more effective here, since the added memory latency in TP is better masked by FlashAttention’s extended compute phase.
>
>
> These points will be clarified in Section 6.3 of the revised manuscript.
>
>
> ### **Q1:** Fig. 3(b) Model Status
> All analyses in Figure 3(a) and 3(b) use an **off-the-shelf, pre-trained GPT-2** checkpoint—**no fine-tuning** was performed. Specifically, we load the HuggingFace GPT-2 (774 M) checkpoint and run it on WikiText-2, PTB, BookCorpus, and CC-News to study its internal behavior in a stabilized setting. As noted in Appendix A.1, no additional training or task-specific fine-tuning occurs for these experiments.
>
>
> ### **Q2:** Fig. 3(b) Ablation with Other Layers
> We thank the reviewer for the suggestion. When we reuse the attention of later layers instead of and along with the first one, we observe the perplexity degradation even compared to the GPT-2 baseline.
>
> **Latest‐Layer Variant (Ablation 1 in Appendix D.1):** we replace the first‐layer signal with each block’s own MHA output. This degrades validation perplexity to 21.34 versus 17.75 for the GPT-2 baseline, confirming that the first attention is uniquely informative.
>
> **Layer‐2&3:** Under fast‐training (“Cramming”) conditions at 48 layers (like Figure 9 - Middle), using the second and third‐layer output atop the first one yields an MLM loss of 2.39, 2.58, respectively, worse than the Pre-LN baseline (2.32) and both FAL (2.22) and FAL+ (2.17). This implies that **additionally injecting less dominant signals along with the first one adversely affects understanding of input during training**.
>
> We will include these ablation study results in Appendix D.1.
>
> Note, in addition to the perplexity degradation, injecting later-layer signals also necessitates additional intra-block all-reduce operations --- this is directly counter to FAL's design goal to halve the communication overhead.

---

### Official Review · Reviewer_eLP5 · 2025-07-06

**Clarity:** 4
**Significance:** 3
**Originality:** 3
**Rating:** 4
**Confidence:** 3

**Summary:**

This paper proposes **FAL** (First Attentions Last), an architecture that removes each block’s
MHA-MLP link and instead feeds the first attention output, which has been normalized, to all
later MLPs. This halves per-block all-reduce traffic in Tensor Parallelism, since the all-reduce
communication between each MHA-MLP has been replaced. This enables parallel MHA and
MLP execution and up to 44\% faster multi-GPU training (\(\le\!1.18\times\) single-GPU
throughput). A variant **FAL+** augments rather than replaces the link, yielding even lower
perplexity. Overall, the approach cuts communication, improves efficiency, and slightly
improves or preserves language-model quality versus GPT-2 baselines.

**Questions:**

- The first attention tensor is reused in every subsequent block; how much additional activation
memory does this introduce for long sequences and deep models? In the paper, minimizing
overhead (including memory usage) is mentioned as one advantage, but it would be better if it
is tested.
- Has this model been tested with an even-larger scale LLM-models or vision-language
models, in which case the "first impression matters" or "similarity stability" may fail? It is good
that in motivation, similarity analysis and connection ablation hold for this case, but it would be
better if the generality to other cases are tested.
- During instruction tuning or RLHF, where later layers often adapt more than early ones, does
repeatedly injecting a frozen first-layer attention harm adaptation or cause catastrophic
forgetting? It would be better if it is tested, and consider whether dynamic first-layer attention
will work better

**Ethical Concerns:**

["NO or VERY MINOR ethics concerns only"]

**Limitations:**

Here is one suggestion to improve the insight of the paper. The idea of this paper is impressive:
utilizing the first attention to later MHA-MLP connections to get the intermediate activation can
greatly reduce overhead, including synchronization waiting and memory usage. However, the
reason of the potential improvement should also be conjectured, or even proved. Intuitively, we
can interpret the improvement by regarding original Pre-LN transformer as a recurrent neural
network, where the output from the previous layer is passed to the next one in a sequence.
However, FAL, proposed in this paper, is similar to attention mechanism, which imitates the
transformer to relate MHA-MLP attention with first attention feature (not relate to other
attention layers due to "first impression matters" in the paper). Therefore, we may conjecture
that: the reason why FAL outperforms traditional methods in some cases may follow a similar
reason why transformer generally outperforms recurrent neural networks in. If possible, this
idea can be evaluated through experiments or reasoning.

**Quality:**

3

**Strengths And Weaknesses:**

**Strengths**
- *Reasonable motivation.* After realizing single MHA output change does not influence the
MLP input greatly, and the output from the first attention layer dominates instead, the need of
all-reduce in each block may be unnecessary.
- *Elegant insight.* Re-uses the first‐layer attention as a cheap and effective substitute for
every MHA-MLP link.
- *Practical speedups.* Reports up to 44\% end-to-end training time reduction on 4×PCIe
GPUs and $1.18\times$ single-GPU throughput, with the same PyTorch code path.
- *Quality preserved.* Perplexity drops slightly and zero-shot SuperGLUE scores hold or rise,
especially with the FAL$^{+}$ variant.


**Weaknesses**
- *Limited task breadth.* Experiments centre on GPT-2 style LM pre-training and SuperGLUE;
vision or multilingual models are untested.
- *Statistical rigor.* All metrics come from single runs, and the variance of the performances are
not tested.
- *Memory analysis absent.* Holding the first attention tensor through $N$ blocks may raise
memory needed to cache the features, but this is not measured nor discussed.
- *Appendix deficiency.* Some contents, like analyzing the limitations, are not included in the
main contents, which would be better to be mentioned in some parts.
- *Downstream robustness.* No fine-tune tasks (instruction tuning, RLHF, retrieval‐augmented
inference) are benchmarked, so practical quality impact is under-explored.
- *Model scale ceiling.* Experiments stop at $8.3$ B parameters and $\le 60$ layers; effects on
truly large LLMs ($\ge 50$ B, hundreds of layers) remain untested.

---

> ### Author Rebuttal · Authors · 2025-07-31
>
> We appreciate your valuable feedback.
>
> ### **W1&Q2:** Limited Task Breadth
> To further demonstrate its cross-model applicability, we provide the following evidence:
>
>
> - For a **vision model (ViT)**, as in Appendix C.2 and E.2, we observe that the similarity of MLP inputs (0.9409 on average) remains consistently higher than that of MHA outputs (0.7859) or MLP outputs (0.7637). Additionally, the gradient magnitude of the first attention layer is 5.42× larger than that of other layers. Leveraging this signal, FAL+ achieves a +1.4% accuracy gain over the ViT baseline.
>
>
> - We have also newly extended our motivational analyses to larger-scale models, including **CodeLLaMA-34B (LLM)** and **VisionLLaMA-11B (vision-language, multilingual)**, further supporting the generality of our findings across modalities and scale. **The results shown in the below tables align with our motivation analyses conducted on GPT-2 and ViT. This 1) demonstrates high similarity in MLP inputs and 2) highlighting the pivotal impact of the first attention on overall model quality**. We will incorporate these new findings into the paper.
>
>
>
> >Similarity Analysis Results (CKA Similarity ± std)
> >
>
> | Activation | LLaMA2-7B | CodeLLaMA-34B | VisionLLaMA-11B |
> |------------|----------|---------------|-----------------|
> | **Attn Out** | 0.60 ±0.14 | 0.66 ±0.13 | 0.84 ±0.10 |
> | **MLP In**   | 0.98 ±0.03 | 0.99 ±0.03 | 0.98 ±0.02 |
> | **MLP Out**  | 0.70 ±0.23 | 0.80 ±0.18 | 0.80 ±0.20 |
>
> >Layer Ablation vs Connection Ablation (Validation PPL)
> >
>
> | Setting | LLaMA2-7B | CodeLLaMA-34B | VisionLLaMA-11B |
> |---------|----------|---------------|-----------------|
> | **Original** | 7.39 | 1.80 | 25.12 |
> | **Remove Layer** | 5339.99 | 1484.52 | 3995.84 |
> | **Remove Conn.** | 892.12 | 37.06 | 504.72 |
>
> >Gradient Analysis (Gradient L1 Norm)
> >
>
> | Block | LLaMA2-7B | CodeLLaMA-34B |
> |-------|---------:|--------------:|
> | **1st** | 505.99 | 321.65 |
> | **2~End avg ±std** | 85.90 ±102.00 | 46.24 ±60.39 |
> | **Ratio (1st/avg)** | 5.9× | 7.0× |
> >Layer Ablation per Block (Validation PPL)
> >
> | Block            | LLaMA2-7B | CodeLLaMA-34B | VisionLLaMA-11B |
> |------------------|---------:|--------------:|----------------:|
> | **1st**          | 34.37 | 4.56 | 40.94 |
> | **2~End avg ±std** | 4.37 ±1.37 | 1.81 ±0.01 | 24.75 ±1.25 |
> | **Ratio (1st/avg)** | 7.9× | 2.5× | 1.7× |
>
>
> - Note, even for the model variants, such as GQA and MoE-based attention, FAL and FAL+ maintain consistent loss improvements over the baseline (see Appendix E.1).
>
>
>
> ### **W2:** Statistical Rigor
> **Performance Measurements:** All reported speedups are averaged **over 50 runs** after an initial warm-up period to stabilize thermal and caching effects, on four representative platforms.  Across these settings, the **relative deviation of measured training/inference times is below 0.01%**, confirming that our reported gains are robust and not due to measurement noise. We will include error bars in the final paper to reflect the variance across multiple runs.
>
>
> **Model Quality:** Although we have not conducted exhaustive seed swaps due to the limited GPU resources available, we believe the model quality results will be consistent --- **pretraining and zero-shot performance evaluation across three distinct corpora---OpenWebText, SuperGLUE, and Pile—exhibit consistent perplexity and accuracy patterns in every case**.
>
> If we get access to more GPU resources in near future, we will conduct the experiments with more random seeds to further ensure the consistent conclusions --- we have tried to get the access during the rebuttal period, but we failed due to contentions.
>
>
>
> ### **W3&Q1:** Memory Analysis Absent.
> FAL does not incur any additional memory overhead during training --- it rather slightly reduces total memory usage compared to the Pre-LN Transformers.
>
>
> **No Extra Activation Buffers:** To prevent storing additional activations—which would be required if the original MHA output were transformed (e.g., via multiplication, addition, or compression)—we **reposition** the first layer’s Layer Normalization (from MLP’s input to MHA’s output). This enables later blocks to reuse the output of the first layer's Layer Normalization without any recomputation as well as memory overhead. Note, every MHA output (including the first layer’s) is already cached during the forward pass to enable gradient computation.
>
>
> **Fewer LayerNorms → Slightly Reduced Memory:** FAL **reduces** total memory usage by approximately 1% compared to the baseline, primarily by reducing the number of LayerNorm operations and associated intermediate activations. Instead of applying a fresh LayerNorm in every block, FAL normalizes the first-attention output only once (in block 1) and reuses the result across all blocks.
>
>
>
> ### **W4:** Appendix Deficiency
> We will add the concise summarization of the Appendix contents (such as key limitations) in the main text.
>
>
>
> ### **W5&Q3:** Downstream Robustness
> As the reviewer suggested, we empirically examined how injecting the first attention signal during instruction tuning affects the trade-off between adaptation and stability — note, given limited rebuttal period (and resources), we have not evaluated RLHF setting in this rebuttal due to its high memory and time requirements. We fine-tune both GPT-2 1.5B and FAL+ 1.5B on the **Alpaca instruction tuning dataset** using four different learning rates (1e-5 to 1e-2). We report:
>
>
> **Trained perplexity on Alpaca**, reflecting adaptation, and
>
>
> **Δ Validation perplexity on OpenWebText**, reflecting forgetting.
>
> | Model        | LR   | Δ Val PPL | Trained PPL |
> |--------------|-----|---------:|-----------:|
> | **FAL+‐1.5B** | 1e-5 | **0.00** | **27.97** |
> |              | 1e-4 | **0.00** | **27.98** |
> |              | 1e-3 | **-0.01** | **27.22** |
> |              | 1e-2 | **0.61** | 5.76 |
> | **GPT2‐1.5B** | 1e-5 | 0.01 | 30.93 |
> |              | 1e-4 | 0.00 | 30.92 |
> |              | 1e-3 | 0.01 | 28.10 |
> |              | 1e-2 | 1.39 | **4.83** |
>
>
> **FAL+ consistently preserves pretraining knowledge better than GPT-2.** Across all learning rates, FAL+ shows lower or equal ΔPPL on OpenWebText compared to GPT-2, indicating stronger stability. At LR=1e-3, FAL+ even shows a slight improvement in validation PPL (-0.01), suggesting positive regularization rather than forgetting.
> **FAL+ also achieves better adaptation when forgetting is minimal.**  While maintaining zero or negative ΔPPL, FAL+ attains the lowest trained PPL (27.22) at LR=1e-3, demonstrating high adaptability with minimal forgetting. In contrast, GPT-2 reaches a lower trained PPL (4.83) at LR=1e-2 only by severely compromising original domain knowledge (ΔPPL = 1.39).
> We agree with the reviewer that **dynamically injecting the first attention** (e.g., through gating or reweighting) is an interesting direction. However, such mechanisms would require **storing additional intermediate activations** and introducing **block-specific parameters**, resulting in **non-trivial memory and compute overhead**. Balancing this added complexity against adaptability gains would require careful exploration, which we leave for future work.
>
>
>
> ### **W6:** Model Scale Ceiling
> While our main experiments are done on a relatively small scale compared to SOTA LLMs, both **efficiency gains and quality preservation/improvement are scalable to larger models**. For the details, please see our response to **W2&Limitations** of **Reviewer R6Kr**. Note we have conducted experiments with 34B CodeLLaMA instead of a 50B model due to the limited resources. Our request (to run the experiments of 50B model on 8 H100 GPUs) is waiting in the queue of the cloud server. If we finish executing the experiments, we will update the results in the paper.
>
> ### **Response to Suggestion:** How FAL Mitigates Information Dilution in Deep Transformers
> We thank the reviewer for this insightful framing and let us answer your suggestion.
>
> **Sequential “RNN‐like” Information Dilution:** In a standard Pre-LN Transformer, each block processes only its immediate predecessor’s output, analogous to unrolling an RNN over depth. As depth grows, early‐layer signals must traverse many transformations and risk dilution or loss—much like long RNN sequences forget initial states.
>
>
> **FAL as a Persistent “First-Impression” Skip:** FAL breaks pure sequential dependence by feeding the first-layer attention output directly into every later block, akin to self-attention’s ability to attend globally. Concretely, we normalize the first MHA output once and then add it to each block’s input:
> MLP_input_i = LN(X_i) + LN(MHA_1(LN(X_1)))
>
>
> **Preventing Signal Dilution in Deep Transformers:** Residual connections alone cannot prevent the first signal from fading: activation variance accumulates layer by layer (Xiong, Ruibin, et al., 2020), eventually washing out early-layer information —much like how self-attention’s O(n²) interactions dilute key dependencies over very long sequences (Yao, Shunyu, et al., 2021).  FAL sidesteps this pitfall by reusing normalized first-attention tensor at each block, reinforcing the most salient initial context without extra overhead—echoing cognitive insights that rethinking a first impression can lead to better decisions in deep reasoning (Wason et al., 1974).
>
>  - Xiong, Ruibin, et al. "On layer normalization in the transformer architecture." International conference on machine learning. PMLR, 2020.
> - Yao, Shunyu, et al. "Self-attention networks can process bounded hierarchical languages." arXiv preprint arXiv:2105.11115 (2021).
> - Wason, Peter C., and J. St BT Evans. "Dual processes in reasoning?." Cognition 3.2 (1974): 141-154.
>
> **Empirical Weighting Ratios:** Analysis of the learned LN‐affine parameters shows that blocks weigh the first-attention signal at ratios ranging roughly from 1 : 1 to 1 : 2 compared to the current (block) input, confirming its active, dynamic role across depth. We will include detailed per-layer ratio plots in Appendix D.

---

> > ### Author Response · Authors · 2025-08-05
> > **We sincerely hope our updates reflect your thoughtful feedback**
> >
> > **Dear Reviewer eLP5,**
> >
> > Thank you for your detailed and constructive review. You raised four important points in your questions and suggestions:
> > 1. Task and model generality,
> > 2. Memory overhead for long sequences and deep models,
> > 3. Robustness during instruction tuning or RLHF, and
> > 4. Suggestion: FAL vs. Transformer.
> >
> > We address each below:
> > 1. Our Motivation analyses extend to **larger models** (LLaMA2-7B, CodeLLaMA-34B) and **vision/multimodal tasks** (ViT-B, VisionLLaMA-11B).
> > 2. We clarify that **no additional activation memory is required**: During training, FAL reuses already cached forward tensors and avoids recomputation via a repositioned LayerNorm. Overall, FAL slightly reduces memory, and FAL+ adds only \~1% predictable overhead.
> > 3. Following your suggestion, we tested FAL+ and GPT-2 during **instruction tuning** (Alpaca /w 4 learning rates). FAL+ consistently shows **better stability** (lower or equal ΔPPL) and **strong adaptation** (lower trained PPL) than GPT-2, indicating a **stronger stability–plasticity trade-off**.
> > 4. We believe FAL mitigates signal dilution by persistently injecting the normalized first-attention output into all layers, **serving as a stable global context**. Learned affine weights per-layer confirm this **first-attention signal is actively used across depth**.
> >
> > Your comments prompted several clarifications and extensions, which we have incorporated into the revised manuscript. We have carefully addressed each point, and would greatly appreciate it if you could reconsider your rating should you find the responses satisfactory.
> >
> > **Best regards,**
> >
> > The Authors

---

### Note · Authors · 2025-08-12

We thank all reviewers for their valuable feedback. Our work is a cross-domain study that tackles a system-level bottleneck through an architectural modification. Overall, our work has been recognized for its **reasonable motivation with elegant and intuitive insight** (Reviewers eLP5, R6Kr), **novel solution that is easy to understand** (Reviewers EX1R, R6Kr), and **promising results with practical speedups and improved model quality** (Reviewers eLP5, EX1R, R6Kr).

Using the Official Comments for each reviewer, we have carefully addressed all raised concerns with

extended analyses:
- **Generality (also Scaling)** across model architectures and domains (Llama 2-7B, CodeLlama-34B,  Llama 3.2 Vision-11B, ViT, attention variants (GQA, MoE)),
- **Downstream robustness** via instruction-tuning experiments on the Alpaca dataset,
- **Scaling (quality)** - as depth grows, Pre-LN transformers suffer “RNN-like” information dilution. FAL mitigates this by reintroducing the normalized first-attention output at every block, preserving initial context and explaining quality gains at 36→48→60 layers (and 774M vs. 1.5B).

and clarifications:
- **Scaling (speedup)** - model-independent speedups by halving communication frequency, with gains becoming more significant as the compute/communication gap widens,
- **Statistical rigor** confirming speedups averaged over 50 runs (<0.01% deviation); and consistent quality trends across three corpora (with plans for additional multi-seed experiments),
- **Memory usage** confirming that FAL slightly reduces memory usage, while FAL+ adds negligible overhead,
- **Overlap mechanism** clarifying how FAL issues parallel MHA/MLP kernels (no fusion required) and refining our statement regarding FlashAttention.

We believe these additions strengthen the technical soundness, applicability, and clarity of our contribution, and we hope the addressed points provide further confidence in the significance and practicality of **FAL** and **FAL+** for efficient transformer training. **Please note that, although we addressed all reviewers’ initial comments, there were no follow-up discussions.**

---

### Decision · Program_Chairs · 2025-09-17

**Decision:**

Accept (poster)

**Comment:**

This paper proposes FAL, which greatly saves multi-GPU communication overhead by removing MHA-MLP communication bottlenecks. Reviewers appreciated the careful ablations and clear presentation - there was initially some concern about potential specificity/overfitting to the GPT-2 architecture, but the authors addressed this concern adequately in their rebuttal. Overall, the scientific rigor and practical results makes the paper of interest to the broader community.